# LEARNING TO REJECT LOW-QUALITY EXPLANATIONS VIA USER FEEDBACK

## ABSTRACT

Machine Learning predictors are increasingly being employed in high-stakes applications such as credit scoring. Explanations help users unpack the reasons behind their predictions, but are not always "high quality". That is, end-users may have difficulty interpreting or believing them, which can complicate trust assessment and downstream decision-making. We argue that *classifiers should have the option to refuse handling inputs whose predictions cannot be explained properly* and introduce a framework for ***learning to reject low-quality explanations*** (LtX) in which predictors are equipped with a *rejector* that evaluates the quality of explanations. In this problem setting, the key challenges are how to properly define and assess explanation quality and how to design a suitable rejector. Focusing on popular attribution techniques, we introduce ULER (User-centric Low-quality Explanation Rejector), which learns a simple rejector from human ratings and per-feature relevance judgments to mirror *human* judgments of explanation quality. Our experiments show that ULER outperforms both state-of-the-art and explanation-aware learning to reject strategies at LtX on eight classification and regression benchmarks and on a new human-annotated dataset, which we publicly release to support future research.

## 1 INTRODUCTION

Machine Learning (ML) predictors are increasingly deployed in *high-stakes* decision-making applications, such as medical diagnosis and credit scoring (Litjens et al., 2017; Pesapane et al., 2018; Gogas and Papadimitriou, 2023). In these domains, incorrect predictions can lead to severe consequences (Kotropoulos and Arce, 2009). To promote trust, *Learning to Reject* (LtR) allows models to defer predictions to human experts if the model has an elevated risk of making a misprediction (Chow, 1970). Traditional LtR approaches typically abstain when the model is uncertain about its prediction or a test example differs substantially from the observed training data (Liu et al., 2020; Ruggieri and Pugnana, 2025).

Currently, LtR neglects a critical aspect of decision-making: *explanation quality* (Kim et al., 2024), cf. Fig. 1 (left). In many applications, it is equally important that models provide clear and convincing explanations for their predictions (Hagos et al., 2022). Without addressing explanation quality, a model might make predictions that cannot be satisfactorily explained. We argue that low-quality explanations can affect trust assessment and downstream decisions (Gilpin et al., 2018; Schneider et al., 2023; Lakkaraju and Bastani, 2020) or induce over-reliance by persuading users to accept incorrect predictions (Joshi et al., 2023; Si et al., 2024; Sieker et al., 2024). As a consequence, we believe models should *offload predictions that they cannot properly explain* to human stakeholders. This ensures that predictions are based on human-validated reasoning and preserves the overall trustworthiness of the system. In high-stakes applications, returning only the prediction is not acceptable when its accompanying explanation is low-quality because explanations are increasingly becoming a legal and regulatory requirement (European Parliament and Council of the European Union). This perspective aligns with the Four Principles of Explainable Artificial Intelligence (Phillips et al., 2021), an official document from the U.S. government, which emphasizes the importance that an AI system recognizes and declares its knowledge limits. According to the authors, "safeguarding answers so that a judgment is not provided when it may be inappropriate to do so" can prevent "misleading, dangerous, or unjust outputs". E.g., consider a general practitioner that uses an AI system to assist in diagnosing malignant melanoma. When examining a suspicious lesion, the AI correctly

Figure 1: **Illustration of** ULER. Learning to Reject (LtR) is unconcerned with the quality of machine explanations (left). ULER *instead addresses Learning to Reject Low-Quality Explanations* (LtX), which requires to reject predictions that cannot be explained properly to stakeholders, improving trust assessment and down-stream decision quality (right).

advises against further action, citing the size of the lesion as a key factor, which is irrelevant in the doctor's opinion. Distrusting the AI's explanation, the doctor decides to proceed with additional examinations, resulting in unnecessary costs and delays.

To formalize this notion, we introduce the ***Learning to Reject Low-Quality Explanations*** (LtX) problem where a model should abstain from making a prediction when it can only provide an unsatisfactory explanation from the user's perspective, cf. Fig. 1 (right). This is a challenging problem that current techniques cannot adequately address. On the one hand, LtR focuses only on prediction quality but just because a model can offer a correct prediction does not imply it can offer an acceptable explanation for it. On the other hand, existing metrics for evaluating explanations do so on the basis of properties of the model. Consequently, these may not align with a human's assessment of the quality of the explanation.

To address the LtX problem, we propose ULER (User-centric Low-quality Explanation Rejector) to train a novel type of rejector to assess the quality of an explanation from a user's perspective. It does so by leveraging expert annotations comprising quality judgments and optionally per-feature relevance judgments. ULER consists of two main steps. First, to avoid having to collect a large num- ber number of explanation judgments, we apply a novel quality-aware augmentation strategy that exploits the human annotations to augment the training set. Second, we fit the rejector to evaluate the explanations' quality using the augmented quality judgment labels. Empirically, we demonstrate that ULER outperforms many popular LtR strategies as well as approaches to estimate the quality of the explanation on both the machine and human side. Finally, to show the effectiveness of ULER on real data, we collected a new larger-scale dataset of human-annotated machine explanations which will make publicly available.

**Contributions**: Summarizing, we: (*i*) Introduce the problem of *learning to reject low-quality explanations* (LtX), filling a significant gap in current LtR strategies, which ignore explanation quality altogether. (*ii*) Design ULER, a rejector that uses modest amounts of human annotations – including explanation ratings and per-feature relevance judgments – to learn an effective rejection policy. (*iii*) Empirically evaluate ULER on both popular data sets and on a novel human-annotated task collected specifically for this work, showcasing its benefits over standard LtR and state-of-the-art explanation quality metrics. (*iv*) Provide the first larger-scale (1050 examples, 5 annotations each) data set of human-annotated explanations as well as a template for running the associated collection campaign.

## 2 PRELIMINARIES

We describe the setup followed throughout the paper. We consider a *predictor $f$* that maps inputs $\boldsymbol{x} \in \mathcal{X}$ to a target value $f(\boldsymbol{x}) \in \mathcal{Y}$. Here, $\mathcal{X}$ is a $d$-dimensional feature space and $\mathcal{Y}$ a discrete ($\mathcal{Y} = \{1, \ldots, C\}$) or continuous ($\mathcal{Y} = \mathbb{R}$) target space. When the target is discrete, we view the predictor as a probabilistic *classifier* that assigns a predictive distribution $P(Y|X = \boldsymbol{x})$ to each input $\boldsymbol{x}$; predictions are obtained via MAP inference, that is $f(\boldsymbol{x}) = \arg\max_{c \in \mathcal{Y}} P(Y = c|\boldsymbol{x})$ (Koller and Friedman, 2009). When the target is continuous, we view it as a *regressor* $f(\boldsymbol{x}) = \mathbb{E}[Y|X = \boldsymbol{x}]$.

In the following, we assume the predictor is paired with an *explainer $e$* which produces a local explanation $\boldsymbol{z} = e(f, \boldsymbol{x})$ of individual prediction $f(\boldsymbol{x})$. Specifically, we focus on *feature importance* explanations, perhaps the most well-known and widespread class of explanations (Guidotti et al., 2018; Ribeiro et al., 2016; Lundberg and Lee, 2017; Ignatiev et al., 2019; Montavon et al., 2017;

Mothilal et al., 2020; Selvaraju et al., 2020). These associate a *relevance score* $z_i \in \mathbb{R}$ to each input feature $x_i$ that quantifies its relative contribution for the prediction. For example, in loan approval, $\boldsymbol{z}$ might indicate that an application $\boldsymbol{x}$ was rejected (*i.e.*, $f(\boldsymbol{x}) = 0$) because a specific feature $x_{\texttt{income}}$, which is too low, "votes" against approval by assigning it a negative value (*i.e.*, $z_{\texttt{income}} < 0$). We refer to the pair $(f(\boldsymbol{x}), \boldsymbol{z})$ as the model *output*, since each prediction $f(\boldsymbol{x})$ is returned to the user along with its corresponding explanation $\boldsymbol{z}$.

**Learning to reject**. To promote trust, a ***Learning to Reject*** (LtR) model combines a predictor $f$ with a *rejector* $r$. The role of the rejector is to offload difficult predictions to a human expert (Franc et al., 2023; Pugnana et al., 2024). Formally, it does so by extending the target space $\mathcal{Y}$ to include an additional symbol Ⓡ indicating the model abstains from making a prediction (Stefano et al., 2000; Cortes et al., 2016a). Two classes of rejection strategies have been studied in the literature. *Ambiguity rejection* occurs when the predictor $f$ is too uncertain about a particular input $\boldsymbol{x}$, *e.g.*, due to class overlap or poor choice of the predictor's hypothesis space (Pugnana and Ruggieri, 2023a; Perini and Davis, 2023). *Novelty rejection* checks if $\boldsymbol{x}$ falls in a region where there is little or no training data (Van der Plas et al., 2021). Although existing rejection strategies improve the model's reliability (Geifman and El-Yaniv, 2017), they focus solely on predictor's performance (Hendrickx et al., 2024) and ignore cases where the explanations themselves are unsatisfactory to the user.

**Metrics of Explanation Quality.** Since explanation quality admits multiple interpretations, numerous metrics have been proposed to evaluate it (Chen et al., 2022). Most of them depend solely on the relationship between the explanation and the predictor and, as such, can be computed accurately using information gathered during inference and/or training. For example, ***faithfulness*** (Mothilal et al., 2021; Azzolin et al., 2025) measures whether an explanation accurately reflects the model's reasoning process, and it is typically computed by assessing whether the features with high relevance are sufficient and necessary for the prediction. Another key metric is ***stability*** (Slack et al., 2021), which measures the degree to which different (possibly conflicting) explanations can be provided for a given prediction. Despite their utility, recent works (Kazmierczak et al., 2024; Colin et al., 2022) have shown that *these metrics do not align with human judgment*, highlighting the need for alternatives. An exception is PASTA, a novel perceptual quality metric that mimics human preferences across multiple dimensions (Kazmierczak et al., 2024) and that we compare against in our experiments (Section 4). Appendix B provides a deeper discussion of these metrics. Although several metrics of explanation quality exist, none have been integrated into rejection strategies to guide the rejector's decisions. Next, we address this gap by introducing a novel framework that incorporates user-perceived explanation quality into the rejection process.

## 3 LEARNING TO REJECT LOW-QUALITY EXPLANATIONS

We introduce the *Learning to Reject Low-Quality Explanations* (LtX) problem where a rejector acts as a filter based on the user-perceived explanation quality (Hoffman et al., 2018; Hsiao et al., 2021). **Specifically, explanation quality reflects two complementary dimensions:** *plausibility*, **meaning that the relevance scores should align with the user's domain knowledge, and** *interpretability*, **meaning that the explanation should be understandable to the user.** Consequently, the rejector in this setting operates on $\boldsymbol{z}$ as opposed to $f(\boldsymbol{x})$ or $\boldsymbol{x}$ as in a standard LtR setting. Formally, a model with reject option in the LtX setting is defined as follows.

**Definition 1.** *An LtX model $m$ consists of three components: a predictor $f$, an explainer $e$ and a rejector $r$. Given (test) instance $\boldsymbol{x}$, $m$ computes $f(\boldsymbol{x})$ and corresponding explanation $e(f, \boldsymbol{x})$. Then, $m$ applies the rejector $r$ to $e(f, \boldsymbol{x})$ to assign a score representing the quality of the explanation $\boldsymbol{z}$ with lower scores being associated with worse explanations. If the score is below a threshold $\tau$, the model abstains from providing the prediction and the corresponding explanation to the user. Formally, $m$ is defined as:*

$$m_{(f,e,r)}(\boldsymbol{x}) = \begin{cases} \text{Ⓡ} & \text{if } r(\boldsymbol{z}) < \tau \\ (f(\boldsymbol{x}), \boldsymbol{z}) & \text{otherwise} \end{cases} \tag{1}$$

Our key contribution is to learn a rejector that abstains when $e$ provides a low quality explanation from the user's perspective. Obtaining such a rejector is challenging for three reasons. First, LtR strategies determine when the model should abstain based on where the predictor is likely to make a mistake. However, the predictor may still output a correct prediction even when the corresponding

explanation is unreliable, and as such they cannot be used as-is. Second, existing metrics to evaluate explanations focus only on the model's internal functioning and are not able to measure the quality of the explanation from the user's perspective, as we will show empirically in Section 4. Third, training a standard LtR model only requires standard supervised dataset consisting of instances and their target values. In contrast, LtX requires human-judgment labels about the explanations of each prediction which are usually not available and may be time-consuming to obtain.

### 3.1 REJECTING LOW-QUALITY EXPLANATIONS WITH ULER

We propose a novel approach for the LtX problem called ULER (User-centric Low-quality Explanation Rejector) that addresses the aforementioned challenges by (*i*) collecting a small set of user annotated explanations, (*ii*) employing a feedback-driven data augmentation strategy, and (*iii*) training a rejector that estimates the user-perceived quality of an explanation. We detail these steps next.

**The rejector's training data.** ULER assumes access to two sources of expert feedback. First, it has a set of explanations and corresponding ***human quality judgments*** denoted by $\mathcal{D} = \{(z_1, y_{z_1}), \ldots, (z_n, y_{z_n})\}$, where $z$ are the explanations, and $y_z \in \{0, 1\}$ their corresponding human quality judgments ($0 =$ low-quality, $1 =$ high-quality)[1]. This feedback is essential for training an LtX rejector that is aligned with expert judgments of explanation quality. Yet, such annotations can be expensive to acquire and therefore typically available in modest amounts (Teso and Kersting, 2019; Kazmierczak et al., 2024).

Second, to avoid having to collect a large annotated dataset, ULER can optionally exploit ***per-feature human labels***. This more detailed source of information allows us to augment the set of quality judgments. The per-feature labels indicate, for each explanation $z$ in $\mathcal{D}$, what relevance scores the user deems incorrect, if any[2] Formally, we indicate as $\mathcal{W}_z$ (resp. $\mathcal{C}_z$) the indices of the features whose relevance the user deems *wrong* (resp. *correct*). Our experiments support the small annotation cost of the augmentation step, as empirically shown in Appendix C.7. In Section 4.2, we show how to design an annotation campaign to obtain both kinds of feedback.

**Augmenting the data**. The ***augmentation step*** works by perturbing each low-quality explanation using a stochastic transformation that leverages the per-feature labels while keeping $y_z$ fixed. We augment only low-quality explanations since the task is typically unbalanced, *i.e.*, we expect most explanations to be high-quality, and having a more-balanced dataset helps learn a better rejector. If explanation $z$ is low-quality, slightly perturbing the features with correct relevance scores should not affect the explanation label. Formally, for each low-quality explanation $z$ we create $K$ new explanations $z_{aug}$ sharing the same human-judgment label $y_z$ as $z_{aug} \sim \mathcal{N}(z, \epsilon_0 s \times \Sigma)$. Here, $\epsilon_0$ is a hyperparameter controlling the overall magnitude of the perturbations, $\Sigma$ is a diagonal matrix whose elements are the per-feature standard deviations across all explanations in $\mathcal{D}$ and is responsible for rescaling perturbations compatibly with the data distribution, and $s$ is a binary vector used to selectively perturb the features in $\mathcal{C}_z$. In practice, the entries of $s$ corresponding to the indices in $\mathcal{C}_z$ are set to 1 and those in $\mathcal{W}_z$ to 0.

**Learning the rejector.** The rejector is defined by a binary classifier $r$ and a threshold $\tau$. ULER trains the former on the augmented data $\mathcal{D}_{aug}$. ULER is agnostic to the specific choice of classifier: any model class that associates a score with its prediction is possible. Empirically, we find that simple models (*e.g.*, kernel SVMs (Cortes and Vapnik, 1995)) work well. $\tau$ determines how often a prediction and explanation are offered by $m$. Lower values of $\tau$ mean that $m$ will operate more autonomously (*i.e.*, return more prediction-explanation pairs) albeit with the risk that some explanations are low quality. Higher values mean the model is more cautious and only offers predictions-explanation pairs when its more certain about the quality of the explanation but at the cost of offloading more decisions to the user. Hence, this value should be carefully tuned, *e.g.*, on validation to navigate this tradeoff. Two natural strategies are to set $\tau$ such that (i) it achieves a spe-

---

[1]In practice, one has some flexibility about how to collect these labels. E.g., in our user study, we used a 5-point Likert scale and transformed these scores into binary labels.

[2]**In high-dimensional domains, obtaining per-feature human labels can be made cognitively affordable by displaying only a limited number of top-ranked features (*i.e.*, those with the highest relevance scores) which users can reasonably assess. In practice, users are expected to flag either (*i*) features among the presented one whose scores they believe are incorrect, or (*ii*) additional features not shown but which they would expect to have significant importance.**

cific rejection rate on the validation data (*e.g.*, one aligned with a user's capacity to make decisions) or (ii) its rejection rate is equal to the proportion of low-quality explanations in the training set.

## 3.2 BENEFITS AND LIMITATIONS

ULER is designed to identify and offload predictions associated with unsatisfactory explanations, as doing so is crucial for ensuring an accurate decision making. **However, if the goal is also to improve predictive performance, ULER can be combined with state-of-the-art LtR strategies specifically developed for this purpose.** One limitation of ULER is that, just like PASTA (Kazmierczak et al., 2024), it relies on high-quality human annotations. We argue that this is necessary in high-stakes applications, but also that good annotations are likely to be available anyhow as in these settings expert users *have* to oversee machine decisions at all times (Hoffman et al., 2018; Zhou et al., 2021; Lai and Tan, 2019), and can therefore consistently supply high-quality responses. Our experiments in Section 4 indicate that ULER is quite sample efficient, as it outperforms the SOTA while using less than 1000 annotations, and that augmentation boosts the performance of the rejector. Finally, our study focuses on tabular data rather than images or text. Working with a larger number of features may increase the sample complexity of the rejector. A possible solution is to adapt ULER to work in a rich pre-trained embedding space, as done by PASTA.

## 4 EMPIRICAL EVALUATION

Empirically, we address the following research questions: (**Q1**) Does ULER correlate with existing machine-side explanation metrics? (**Q2**) Does ULER reject more low-quality explanations than the competitors? (**Q3**) **(User study)** Is ULER capable of mimicking human judgments?

The Appendix examines two additional questions: Appendix C.6 explores the effect of what information ULER's rejector has access to on its ability to reject low-quality explanations and Appendix C.7 investigates the effect of the data augmentation based on per-feature feedback on its performance. *Our code is available in the Supplementary Material and will be published upon acceptance*.

**Competitors**. We compare ULER against *eight* representative rejection strategies from two groups: (*i*) standard LtR strategies, and (*ii*) explanation-aware strategies. All strategies yield a score for each input; the $\rho_\%$ inputs with the lowest score are rejected, where $\rho_\%$ is the *rejection rate*.

We consider *three standard LtR strategies* that target improving predictive performance on those examples for which the models offers a prediction. RandRej is a baseline that assigns a random score to each input. NovRej$_X$ rejects inputs based on their novelty (Sun et al., 2022): it first computes their distance to the $k$-th nearest training instances and converts these into scores using a monotonically decreasing function, *e.g.*, $1/(1+x)$, such that farthest inputs get lower scores. PredAmb uses prediction's confidence as score (Hendrickx et al., 2024). For binary classification tasks, confidence is computed as the margin of the class probabilities $|P(Y = 1|\boldsymbol{x}) - P(Y = 0|\boldsymbol{x})|$ (Perini and Davis, 2023). For regression tasks, the conditional variance for each input is computed and then the score is obtained applying a monotonically decreasing function, *e.g.*, $1/(1+x)$, such that higher-variance predictions obtain lower scores (Zaoui et al., 2020).

We consider *five novel but natural explanation-aware strategies*. Three leverage machine-side explanation metrics as scores, one for each category in Chen et al. (2022). Specifically, StabRej looks at the stability of the explanation (Mothilal et al., 2021), measuring the similarity among the different explanations that can be generated for the same prediction. FaithRej assesses the faithfulness (Azzolin et al., 2025) of an explanation by measuring how well the explanation identifies features that are truly causally relevant for the prediction. ComplRej measures the complexity (Bhatt et al., 2020) of an explanation *i.e.*, the cognitive load it enforces on a user; since low-complexity explanations are preferred, the score is obtained applying a monotonically decreasing transformation, *e.g.*, $1/(1+x)$, to the metric value. PASTARej uses an adaptation of the state-of-the-art human-side PASTA-metric to score each explanation (Kazmierczak et al., 2024). Since our focus is on tabular data, we drop the embedding network and fit only the scoring network using the explanations as input to learn the human-judgment. **Importantly, our approach fundamentally differs from PASTARej, as it is specifically designed to detect and reject low-quality explanations. While PASTA provides a human-judgment-based metric to score explanations, our method**

**introduces a feedback-aware augmentation strategy for each dataset, enabling the rejector to effectively learn to discriminate between high- and low-quality explanations.** Full details on all metrics are provided in Appendix B. Finally, $\texttt{NovRej}_Z$ mirrors $\texttt{NovRej}_X$ but works in the explanation space, testing whether the perceived low-quality explanations correspond to outlier explanations.

**Evaluation metrics**. $\texttt{ULER}$ aims to capture human judgments of explanation quality, which recent works have shown to be misaligned with existing machine-side metrics (Kazmierczak et al., 2024; Colin et al., 2022). Therefore, to examine whether $\texttt{ULER}$ captures information that existing metris do not, we compute the correlation between the scores computed by $\texttt{ULER}$'s rejector and three existing machine-side metrics: faithfulness, stability, and complexity (see Appendix B.1 for full details). We use the Spearman coefficient as it is sensitive to all monotonic relationships, even non-linear ones (Kendall, 1949).

Q2 and Q3 evaluate the competitors' ability to reject low-quality explanations. Ideally, a user wants to receive only predictions accompanied by high-quality explanations. A good rejector should therefore minimize the number of low-quality explanations it shows to the user (*accepted set*), and maximize the ones for which it abstains (*rejected set*). Thus, we report the percentage of low-quality explanations in the accepted and rejected sets when varying the rejection rate. Moreover, we measure the rejector's ability to rank low-quality explanations below high-quality ones, making them more likely to be rejected, by reporting the AUROC, which is standard in novelty rejection (Sun et al., 2022; Liang et al., 2018).

**Setup**. We employ the following procedure: for each dataset, we (*i*) split $\mathcal{D}$ into $\mathcal{D}_{train}$, $\mathcal{D}_{val}$ and $\mathcal{D}_{test}$ (70%/10%/20%), (*ii*) fit the rejectors on $\mathcal{D}_{train}$ and optimize their hyperparameters on $\mathcal{D}_{val}$, (*iii*) vary the rejection rate $\rho_\%$ from 1% to 25%, and (*iv*) compute the metrics outlined in the previous paragraph on $\mathcal{D}_{test}$. To improve robustness, we repeat steps (*i*)–(*iv*) 10 times and report the average results. All experiments were implemented in Python and executed on an Intel i7-12700 machine with 64 GB RAM. The experiments required approximately two days to complete.

**Model selection**. All explanations are computed using *KernelSHAP* (Lundberg and Lee, 2017) with 100 samples and the predictor's training set as background. We choose *KernelSHAP* as it is one of the most well-known and widely used explainers (Saarela and Podgorelec, 2024). **To further support our findings, we also include results using *LIME* (Ribeiro et al., 2016) in Appendix C.4.** For $\texttt{ULER}$, we train an $\texttt{SVM}$ to assess explanation quality. As mentioned, we optimize $\texttt{ULER}$'s and the competitors' hyperparameters via grid search on $\mathcal{D}_{val}$, see Appendix C.3 for details.

## 4.1 Q1 AND Q2: BENCHMARK DATASETS

**Datasets**. We evaluate all competitors on *eight* widely used benchmarks datasets (Kelly et al., 2023) using simulated human judgments. Since our approach works for any type of prediction function, we select four classification tasks and four regression tasks covering several application domains, including healthcare (*parkinson*), economics (*creditcard*, *adult*), law (*compas*), etc. (*wine*, *bike*, *power*, *churn*). Full details about the datasets are provided in Appendix C.1.

**Simulating human judgments**. We simulate human quality judgments $Y_Z$ and identify features with incorrect relevance scores using a large language model (Llama-3.1-8B-Instruct). Following Domnich et al. (2025), we carefully crafted a prompt that (i) defines the evaluation task, (ii) introduces the structure and meaning of SHAP explanations, and (iii) specifies the expected output format. The LLM was asked to assess the quality of each explanation and identify the features with incorrect relevance scores. Appendix C.2 shows the specific prompt used to obtain the labels. **Additionally, Appendix C.5 evaluates the ability of $\texttt{ULER}$ and all baselines to reject low-quality explanations when the simulated human judgments are generated using a ML oracle.**

**(Q1) Correlation analysis with machine-side metrics.** Table 1 reports each dataset's average Spearman coefficient ($\pm$ std) for each machine-side metric. We would expect correlations that are low in magnitude if $\texttt{ULER}$ captures information that existing metrics do not. With a small number of exceptions, we observe that indeed $\texttt{ULER}$'s scores are not strongly correlated with those of the existing machine metrics as it achieves a correlation $> 0.5$ or $< -0.5$ only three times with faithfulness and once each with stability and complexity. These low correlations confirm that $\texttt{ULER}$ captures information orthogonal to these machine-side metrics. Importantly, repeating the experiment with

Table 1: ULER **is not strongly correlated with existing machine-side metrics.** Average Spearman correlation coefficient ($\pm$ std) between ULER and each machine-side metric across the eight benchmark datasets considered.

|  | faithfulness | stability | complexity |
|---|---|---|---|
| compas | $0.03 \pm 0.11$ | $-0.04 \pm 0.08$ | $0.04 \pm 0.07$ |
| creditcard | $0.05 \pm 0.06$ | $0.76 \pm 0.01$ | $0.66 \pm 0.03$ |
| adult | $0.71 \pm 0.02$ | $-0.25 \pm 0.02$ | $0.24 \pm 0.03$ |
| churn | $0.71 \pm 0.08$ | $0.18 \pm 0.06$ | $-0.22 \pm 0.08$ |
| wine | $-0.14 \pm 0.07$ | $-0.01 \pm 0.07$ | $0.14 \pm 0.05$ |
| parkinson | $0.05 \pm 0.07$ | $-0.05 \pm 0.06$ | $0.08 \pm 0.07$ |
| power | $-0.54 \pm 0.09$ | $-0.01 \pm 0.02$ | $-0.07 \pm 0.06$ |
| bike | $-0.02 \pm 0.04$ | $-0.04 \pm 0.03$ | $-0.05 \pm 0.03$ |

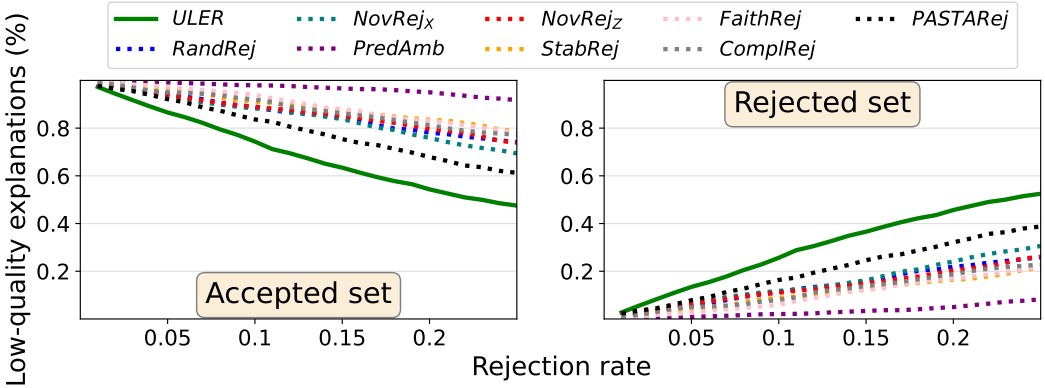

Figure 2: ULER **rejects on average more low-quality explanations than all competitors.** Average percentage of low quality explanations in the accepted and rejected set for all the considered strategies over the 8 datasets for 25 rejection rates $\rho_\%$. For all the considered rejection rates, ULER consistently rejects more low-quality explanations than all competitors.

Table 2: ULER **outperforms the competitors at separating low-quality from high-quality explanations.** Average AUROC for all the rejection strategies over the 8 datasets and its standard deviation. ULER consistently obtains the best results in all datasets.

|  | Classification | | | | Regression | | | |
|---|---|---|---|---|---|---|---|---|
|  | compas | creditcard | adult | churn | wine | parkinson | power | bike |
| ULER | **$0.76 \pm 0.02$** | **$0.56 \pm 0.03$** | **$0.71 \pm 0.03$** | **$0.72 \pm 0.05$** | **$0.80 \pm 0.05$** | **$0.59 \pm 0.08$** | **$0.90 \pm 0.02$** | **$0.78 \pm 0.03$** |
| RandRej | $0.52 \pm 0.04$ | $0.50 \pm 0.03$ | $0.51 \pm 0.05$ | $0.52 \pm 0.08$ | $0.49 \pm 0.09$ | $0.51 \pm 0.09$ | $0.51 \pm 0.1$ | $0.51 \pm 0.09$ |
| PredAmb | $0.42 \pm 0.04$ | $0.50 \pm 0.01$ | $0.35 \pm 0.03$ | $0.28 \pm 0.04$ | $0.56 \pm 0.07$ | $0.49 \pm 0.10$ | $0.51 \pm 0.05$ | $0.57 \pm 0.07$ |
| NovRej$_X$ | $0.70 \pm 0.03$ | $0.48 \pm 0.04$ | $0.50 \pm 0.03$ | $0.57 \pm 0.04$ | $0.65 \pm 0.06$ | $0.56 \pm 0.05$ | $0.28 \pm 0.08$ | $0.62 \pm 0.05$ |
| StabRej | $0.46 \pm 0.04$ | $0.42 \pm 0.03$ | $0.53 \pm 0.03$ | $0.50 \pm 0.0$ | $0.47 \pm 0.06$ | $0.50 \pm 0.08$ | $0.45 \pm 0.06$ | $0.59 \pm 0.09$ |
| FaithRej | $0.39 \pm 0.04$ | $0.49 \pm 0.01$ | $0.33 \pm 0.02$ | $0.27 \pm 0.03$ | $0.52 \pm 0.05$ | $0.49 \pm 0.05$ | $0.65 \pm 0.07$ | $0.53 \pm 0.04$ |
| ComplRej | $0.61 \pm 0.04$ | $0.51 \pm 0.02$ | $0.54 \pm 0.03$ | $0.45 \pm 0.04$ | $0.57 \pm 0.06$ | $0.39 \pm 0.05$ | $0.43 \pm 0.07$ | $0.53 \pm 0.06$ |
| PASTARej | $0.66 \pm 0.14$ | $0.50 \pm 0.05$ | $0.65 \pm 0.04$ | $0.53 \pm 0.07$ | $0.64 \pm 0.15$ | $0.55 \pm 0.06$ | $0.74 \pm 0.20$ | $0.68 \pm 0.10$ |
| NovRej$_Z$ | $0.64 \pm 0.04$ | $0.45 \pm 0.02$ | $0.33 \pm 0.03$ | $0.45 \pm 0.09$ | $0.70 \pm 0.06$ | $0.57 \pm 0.07$ | $0.25 \pm 0.06$ | $0.63 \pm 0.08$ |

Pearson correlation coefficients led to the same qualitative conclusions. For completeness, we also report results on the user study data in Table 8 (Appendix).

**(Q2) Comparison with competitors.** Fig. 2 shows the percentage of low-quality explanations for the accepted and the rejected set as a function of the rejection rate $\rho_\%$ averaged over the eight considered datasets. On average, ULER rejects more low-quality explanations than the competitors: about $10\%$ more than PASTARej, $15\%$ vs NovRej$_X$, $17\%$ vs RandRej and NovRej$_Z$, and over $20\%$ vs FaithRej, StabRej and ComplRej, and PredAmb. Notably, PASTARej, the only competitor that exploits human judgments, outperforms all other baselines, confirming that obtaining such feedback is crucial in the LtX setting.

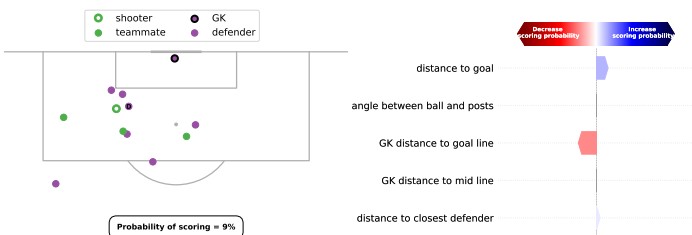

Figure 3: **Image from the user study** illustrating the snapshot (left), the predicted probability of scoring (bottom) and the associated Kernel`SHAP` explanation (right). This suggests that the feature "*distance to goal*" slightly increases the probability, while "*GK distance to goal line*" decreases it.

Table 2 reports the average AUROC per dataset. In all datasets, ULER is better at distinguishing between high- and low-quality explanations than its competitors, with an average improvement of 11% and 17% over the best performing competitors `PASTARej` and `NovRej`$_X$.

### 4.2 Q3: ULER PREDICTS HUMAN JUDGMENTS BETTER THAN THE SOTA

Finally, we collect high-quality human ratings of machine explanation through a large-scale annotation campaign, recruiting users with the crowd-sourcing platform Prolific (https://www.prolific.com),[3] and apply ULER to this dataset to reject low-quality explanations.

Our task was to explain the prediction of an expected goals (xG) model, which values the quality of a scoring opportunity in soccer as the probability that a shot results in a goal (Robberechts et al., 2020). Our choice stems from three considerations. First, Prolific enabled us to recruit subjects that possess the necessary domain expertise to perform the task, cf. Appendix D.3 for our vetting criteria. Second, all instances can be easily visualized, as shown in Fig. 3. Third, this is a real-world task with xG values being shown on TV and used in player recruitment (Graham, 2024).[4] We collected annotations for 1050 explanations from five annotators each, for a total of 5250 annotations.

**Obtaining the explanations**. As a first step, we trained the predictor whose explanations we aim to annotate. Following standard practice in soccer analytics (Robberechts et al., 2020; Robberechts and Davis, 2020), we learned an XGBoost ensemble classifier (Chen and Guestrin, 2016) to estimate the probability of a shot resulting in a goal. The training data consists of 21337 annotated shot events from the 2015-16 season in the top divisions of England, Spain, Germany and France (Statsbomb, 2023). For each shot, the location and the result (goal *vs.* no goal) are recorded. Additionally, a snapshot is available, capturing the locations of the players visible in the broadcast video at the moment the shot is taken, cf. Fig. 3 (left). From this data, we extract features that describe the positions of the shooter, goalkeeper, and nearest defender. Importantly, we include only features that are directly visualizable by the annotators in the snapshot. Explanations are generated on a separate set of 1050 shots from the 2015–16 season of the Italian top division on which the predictor achieves an AUROC of 0.81. All preprocessing and training details are provided in Appendix D.2.

**Obtaining the annotations**. Our goal is to obtain human-judgment labels on the explanation quality and per-feature feedback on the relevance scores. Given that subjective tasks are highly sensitive to interface design (Pommeranz et al., 2012) and question framing (Stalans, 2012), we designed our annotation protocol with the help of a psychologist and conducted several pilot studies to mitigate cognitive biases (Bertrand et al., 2022). Participants ($N = 175$) were recruited via Prolific while annotations were collected through Google Forms. Each participant annotated 30 trials. In each trial, participants were shown a snapshot of a shot and the corresponding prediction and explanation, cf. Fig. 3. The left side shows the position of all involved players and the ball, along with the model's prediction. The right side shows the relevance scores of each feature as arrows indicating whether the feature increases or decreases the predicted probability of scoring. The features were chosen specifically to be easily interpretable and visually grounded, enabling intuitive assessment by the annotators. These were requested to specify how much they agreed with the model's prediction

---

[3]The campaign has received approval from our Research Ethics committee and Privacy office.
[4]The model used in our experiments is not as complex as deployed models.

and, separately, with its explanation using two 5-point Likert-scale questions ($1$ = completely disagree, $5$ = completely agree). Next, they were asked to optionally select individual features they believed were misused in the explanation, *i.e.*, had an incorrect relevance score, via a multiple-choice question. We validated our experimental design by tracking the consistency of individual annotations in two pilot studies: on average, annotators tended to assign consistent scores to the same explanation across repeated trials. Full details about our procedure are provided in Appendix D.3.

**Annotation preprocessing**. To ensure high-quality annotations, we filtered out participants that failed an attention check, rated all explanations the same, or did not flag any as incorrect, leaving us with 149 participants, as well as explanations with low inter-annotator agreement. We aggregated the explanation scores using the average and considered explanations with an average score lower than 3 as low-quality, and the others as high-quality (Joshi et al., 2015; Batterton and Hale, 2017). For feature-level feedback, we marked a relevance score as incorrect if the majority of annotators agreed that the corresponding feature was misused.

**Results.** We evaluate ULER on the collected annotations and compare it against PASTARej, the only baseline that leverages human judgments and emerged as the runner-up in the previous experiments. ULER achieves an AUROC of $0.64 \pm 0.05$, outperforming PASTARej, which scores $0.53 \pm 0.09$. A paired t-test confirms that the difference is statistically significant ($p < 0.01$). These results indicate that learning human-perceived explanation quality is inherently challenging, especially in this subjective task. The overall low performance can be attributed to this increased variability. Additionally, ULER rejects more low-quality explanations than PASTARej in $\sim 84\%$ of the experiments across rejection rates ($\rho_\% \in [1\%, 25\%]$), confirming its superiority.

## 5 RELATED WORK

**Learning to Reject**. The problem of deferring hard decisions has been studied in the context of *learning to reject*, *learning to defer* (Mozannar and Sontag, 2020), *learning under algorithmic triage* (Raghu et al., 2019; Okati et al., 2021), *learning under human assistance* (De et al., 2020; 2021), and *learning to complement* (Bansal et al., 2021); see (Hendrickx et al., 2024) for a recent survey. These approaches all enable the machine to offload certain decisions to a human expert, but differ in what criterion they use. While some strategies entirely rely on the machine's self-assessed uncertainty (Cortes et al., 2016b; Liu et al., 2022; Pugnana and Ruggieri, 2023b), others implement the rejection policy as a machine learning classifier and optimize it for joint team performance (Madras et al., 2018) or learn the classifier and the policy jointly (Wilder et al., 2021). None of them, however, considers the role of explanations in decision making, which we argue is central. Note that ULER is not meant as a replacement for existing strategies, as it has a different goal. On the contrary, it could and should be combined with them to ensure *both* incorrect predictions and unsatisfactory explanations are deferred. We will evaluate this generalization in future work.

**Explainable AI** (XAI) aims at designing mechanisms for properly justifying algorithmic decisions to end-users in non-technical terms (Adadi and Berrada, 2018). We focus on (post-hoc) feature attribution techniques, which highlight what features influenced a prediction the most. Many high profile techniques belong to this group, *e.g.*, LIME (Ribeiro et al., 2016), SHAP (Lipovetsky and Conklin, 2001; Strumbelj and Kononenko, 2010; Štrumbelj and Kononenko, 2014; Datta et al., 2016; Lundberg and Lee, 2017), input gradients (Simonyan et al., 2013; Sundararajan et al., 2017), and formal feature attributions (Yu et al., 2023). **With respect to feature-attribution methods,** ULER is explainer-agnostic, *i.e.*, it can assess the perceived quality of attributions irrespectively of how these are computed. The only work that combines XAI and LtR is (Artelt et al., 2023), which focuses on explaining the reasons behind rejection using counterfactuals, and as such is orthogonal to our work.

**Evaluating explanations**. There is a large body of work on evaluating explanation quality. Most metrics are "machine-side", in that they only consider properties of the model and of how the explanation is computed (*e.g.*, faithfulness, stability, complexity) (Azzolin et al., 2025; Kalousis et al., 2007; Slack et al., 2021; Dasgupta et al., 2022; Alvarez-Melis and Jaakkola, 2018; Chalasani et al., 2020; Nguyen and Martínez, 2020). Our experiments show that these metrics cannot anticipate whether *users* will agree with or believe in a given explanation. In contrast, we learn our rejector to mimic human judgments of explanation quality. Closest to our work is PASTA (Kazmierczak et al., 2024), which however is not designed for rejection and underperforms in our experiments.

## 6 CONCLUSION

We have introduced the problem of *learning to reject low-quality explanations* (LtX) and proposed ULER, a simple yet effective technique for learning a high-quality rejector from a limited amount of expert feedback. Our empirical analysis showcases how, in contrast to other LtR approaches, ULER successfully identifies low-quality explanations in both synthetic and human-annotated tasks. In future work, we will extend our setup to learn the rejector and classifier jointly, so as to optimize their overall performance (De et al., 2020; 2021; Wilder et al., 2021), and look into leveraging ULER's rejector for debiasing confounded ML models by rating their explanations (Teso et al., 2023).

**Reproducibility statement** All details necessary to reproduce our experiments are provided in Section 4, Appendix C, and Appendix D, including full descriptions of the models and datasets. Section 4 presents the overall experimental setup, Appendix C details the hyperparameters and training settings for the simulated setting, and Appendix D reports the specifics of the user study. The benchmark datasets are available online (Kelly et al., 2023). The user study data and the source code will be publicly released upon acceptance.

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

## A  BROADER IMPACT

Rejecting low-quality explanations can be beneficial from at least two perspectives. First, when human involvement is expensive and time-consuming, this reject option serves as an effective mechanism to filter outputs based on human-validated reasoning. Second, since modern decision-making often relies on both predictions and their corresponding explanations, explanation quality becomes critical to prevent harmful decisions.

Our approach contributes to this goal by enhancing trust in the system and supporting human-validated decision-making, ultimately promoting more effective human-AI interaction. Our findings represent an initial step in this direction, showing that our method can reject more low-quality explanations than several existing and adapted learning-to-reject strategies.

## B  EXPLANATION QUALITY METRICS

Explanation quality metrics aim to assess to what extent explanations satisfy the general goal of explaining a decision. These metrics can be broadly categorized into two families (Lopes et al., 2022; Zhou et al., 2021; Vilone and Longo, 2021): *machine-side* and *human-side* metrics. The former focus exclusively on the relationship between the explainer and the predictor, whereas the latter involve human subjects in evaluating the quality of the explanations.

### B.1  MACHINE-SIDE METRICS.

The simplest way to evaluate an explanation is by verifying whether it effectively reveals the predictor's underlying reasoning. Several metrics have been proposed to assess the relationship between explanations and the predictor. Chen et al. (2022) categorize existing machine-side metrics - and provide their mathematical formulations — into three groups: stability, faithfulness, and complexity. We exclude homogeneity from our analysis because it is defined for groups of explanations rather than individual ones.

**Stability** measures the similarity of explanations under changes to the input instance, the training data or the model hyperparameters (Yeh et al., 2019; Alvarez-Melis and Jaakkola, 2018; Ghorbani et al., 2019; Kalousis et al., 2007; Nogueira and Brown, 2016; Mishra et al., 2021). This can be harmful because an attacker can selectively choose explanations based on their (potentially adversarial) interests (Schneider et al., 2023; Bordt et al., 2022). Following Bansal et al. (2020), we define the stability of an explanation as the average similarity across multiple runs of the same explainer, each potentially yielding a different explanation. Formally, given an instance $\boldsymbol{x}$ and prediction $f(\boldsymbol{x})$ with associated explanation $\boldsymbol{z}$, *stability* is defined as:

$$\text{stab}(\boldsymbol{z}) = \mathbb{E}_{\boldsymbol{z}' \sim \mathcal{Z}}\left[Sim(\boldsymbol{z}, \boldsymbol{z}')\right] \tag{2}$$

where $Sim$ is a similarity metric and $\mathcal{Z}$ denotes the space of possible explanations for the given prediction. In practice, we compute stability using the Pearson correlation coefficient as the similarity metric and average it across ten independently generated explanations.

**Faithfulness** measures how accurately an explanation captures the true underlying behavior of the predictor (Bhatt et al., 2020; Alvarez-Melis and Jaakkola, 2018; Rieger et al., 2020; Nguyen and Martínez, 2020; Dasgupta et al., 2022; Kazmierczak et al., 2024). Given an explanation $\boldsymbol{z}$, we define the sets of relevant features $\boldsymbol{z}_{\mathcal{R}} = \{i < d : |\boldsymbol{z}_i| > 0\}$ and irrelevant features $\boldsymbol{z}_{\mathcal{I}} = \{i < d : |\boldsymbol{z}_i| = 0\}$. Intuitively, an explanation is faithful if perturbing irrelevant features causes little to no change in the predictor's output, while perturbing relevant features induces significant changes. Building on (Azzolin et al., 2025), we define *faithfulness* (faith) as the harmonic mean of *sufficiency* (suf) and *necessity* (nec), which estimate the sensitivity of the prediction to perturbations in irrelevant and relevant features, respectively. Formally, given a instance-prediction pair $(\boldsymbol{x}, f(\boldsymbol{x}))$ with associated explanation $\boldsymbol{z}$, and the predictor to be explained $f$, *sufficiency* and *necessity* are defined as:

$$\text{suf}_{d,p_{\mathcal{I}}}(\boldsymbol{z}) = \mathbb{E}_{\boldsymbol{x}' \sim p_{\mathcal{I}}}\left[\Delta_f(\boldsymbol{x}, \boldsymbol{x}')\right] \tag{3}$$

$$\text{nec}_{d,p_{\mathcal{R}}}(\boldsymbol{z}) = \mathbb{E}_{\boldsymbol{x}' \sim p_{\mathcal{R}}}\left[\Delta_f(\boldsymbol{x}, \boldsymbol{x}')\right] \tag{4}$$

where $\Delta_f$ measures prediction change between $\boldsymbol{x}$ and its perturbed version $\boldsymbol{x}'$, and $p_{\mathcal{R}}$ and $p_{\mathcal{I}}$ are interventional distributions that specify how to perturb relevant and irrelevant features, respectively.

Equation 3 and Equation 4 are then normalized to $[0, 1]$ range, the higher the better, via a non-linear transformation *i.e.*, respectively $exp\left(-\operatorname{suf}_{d,p_{\mathcal{I}}}\right)$ and $1 - exp\left(-\operatorname{nec}_{d,p_{\mathcal{R}}}\right)$. Operationally, for a given instance-explanation pair $(\boldsymbol{x}, \boldsymbol{z})$ sampling from $p_{\mathcal{R}}$ ($p_{\mathcal{I}}$) involves perturbing the features in $\boldsymbol{z}_{\mathcal{R}}$ ($\boldsymbol{z}_{\mathcal{I}}$) following Bucila et al. (2006), while keeping the remaining features fixed. Additionally, the prediction change $\Delta_f$ is computed either as the absolute difference in positive class probability for classification tasks, *i.e.*, $|P(Y = 1|\boldsymbol{x}) - P(Y = 1|\boldsymbol{x}')|$, or the absolute prediction difference in regression, *i.e.*, $|f(\boldsymbol{x}) - f(\boldsymbol{x}')|$.

**Complexity** refers to the cognitive burden associated with parsing an explanation (Bhatt et al., 2020; Chalasani et al., 2020; Nguyen and Martínez, 2020). In general, a less complex explanation is easier for a human to understand, making complexity a common proxy for understandability (Cowan, 2001; Molnar, 2020). Following Bhatt et al. (2020), given an instance $\boldsymbol{x}$ with prediction $f(\boldsymbol{x})$ and explanation $\boldsymbol{z}$, we formally define *complexity* as:

$$\operatorname{compl}_{d,p_{\mathcal{I}}} = \mathbb{E}\left[-\ln\left(\overline{\boldsymbol{z}}\right)\right] = -\sum_{i=1}^{d} \overline{\boldsymbol{z}_i} \ln\left(\overline{\boldsymbol{z}_i}\right) \tag{5}$$

where $\overline{\boldsymbol{z}_i}$ is the fractional contribution of feature $i$, *i.e.*, the ratio of its absolute relevance score $|\boldsymbol{z}_i|$ to the sum of all the absolute relevance scores $\sum_{j=1}^{d} |\boldsymbol{z}_j|$.

### B.2 HUMAN-SIDE METRICS

Despite the literature recognizing the importance of human-centered evaluations (Kazmierczak et al., 2024; Vilone and Longo, 2021), only a few metrics have been proposed to evaluate explanations from perspective of a human (Naveed et al., 2024). This gap stems from the inherently subjective nature of human evaluations, which typically makes it challenging to provide a precise mathematical formulation for a metric (Chen et al., 2022). Moreover, there is no consensus in the literature regarding standard criteria for human-side evaluation metrics (Zhou et al., 2021).

**PASTA** uses a model to score each explanation based on how this is perceived by humans (Kazmierczak et al., 2024). The authors first construct a dataset in which users rated several explanations according to four key desiderata: faithfulness, robustness, complexity, and objectivity. Then, the *PASTA-metric* is trained on these ratings to derive a metric value for new explanations. Specifically, this model consists of two main components: an embedding network that leverages a foundation model to generate feature embeddings from the explanations, and a scoring network that employs a linear layer to predict the human ratings based on these embeddings. PASTA is the closest competitor to our work in that it also aims to assess explanations based on human feedback. However, there are three substantial differences with our approach. First, PASTA is designed for image data and relies on an embedding network to create embeddings from this high-dimensional space, whereas we focus on tabular data and learn directly from feature-importance explanations. Second, PASTA does not include a rejection mechanism and always returns a score regardless of quality, while we explicitly aim to develop a reject option based on explanation quality. Third, PASTA seeks to create a dataset-agnostic metric and thus annotates 25 explanations per dataset to encourage generalization. In contrast, we aim to train a dataset-specific rejector and therefore collect 1050 annotations for a single dataset.

**Other human-side metrics.** *Understandability* measures whether an explanation is easy to comprehend for the human (Lopes et al., 2022). The rationale behind this metric is to examine whether the explanations facilitate the user's understanding of the model's decisions (Dieber and Kirrane, 2022). *Plausibility* is high if $\boldsymbol{z}$ matches the ground-truth explanation $\boldsymbol{z}^*$, assuming the latter exists and is unique. Depending on the model's behavior and structure of the underlying learning problem, the model's reasoning may or may not reflect the ground-truth explanation $\boldsymbol{z}^*$. Our approach implicitly addresses both metrics. The user's rating depends on how understandable the explanation is, *i.e.*, users tend to assign low scores to explanations they find difficult to interpret. Furthermore, the per-feature feedback we collect encourages users to identify features that substantially deviate from their expectations, thereby aligning the underlying ground truth.

Table 3: **Datasets' characteristics and predictor's performance.** This table reports the datasets' characteristics (*i.e.*, size of the training set $\#(\mathcal{T})$, number of features $d$, size of the test set $\#(\mathcal{D})$, proportion of low-quality explanations $\gamma$) and the predictor $f$'s performance on the eight benchmark datasets used in the experiments.

| dataset | $\#(\mathcal{T})$ | $d$ | $\#(\mathcal{D})$ | $BACC_f \uparrow$ | $\gamma$ |
|---|---|---|---|---|---|
| compas | 10000 | 12 | 2000 | 0.690 | 0.05 |
| creditcard | 10000 | 23 | 2000 | 0.608 | 0.12 |
| adult | 10000 | 12 | 2000 | 0.757 | 0.02 |
| churn | 1000 | 13 | 1850 | 0.696 | 0.15 |

| dataset | $\#(\mathcal{T})$ | $d$ | $\#(\mathcal{D})$ | $MSE_f \downarrow$ | $\gamma$ |
|---|---|---|---|---|---|
| news | 10000 | 58 | 2000 | 0.009 | 0.48 |
| wine | 1000 | 11 | 2000 | 0.015 | 0.02 |
| parkinson | 1000 | 19 | 2000 | 0.044 | 0.46 |
| appliances | 10000 | 27 | 2000 | 0.010 | 0.32 |

## C  EXPERIMENTS: EXTENDED DETAILS AND RESULTS

### C.1  DATASET CHARACTERISTICS AND PREDICTOR'S PERFORMANCE

Table 3 presents the characteristics of the eight datasets used in the empirical evaluation, along with the performance of the predictor $f$. We report the balanced accuracy (*BACC*) for classification tasks; for regression tasks, we report the mean squared error (*MSE*) after normalizing the target variable to the $[0, 1]$-range. Specifically, the predictor $f$ is trained on a training set $\mathcal{T}$ and evaluated on a test set $\mathcal{D}$. The size of $\mathcal{D}$ is limited because obtaining human-judgment labels on explanation quality is expensive (Kazmierczak et al., 2024). Additionally, the table reports the proportion of low-quality explanations $\gamma$ in $\mathcal{D}$ for each dataset, as determined using the procedure described in Section 4.1.

### C.2  EXAMPLE PROMPT

We use the Llama-3.1-8B-Instruct large language model (LLM) to obtain simulated human quality judgments and to identify features with incorrect relevance scores. Below is the prompt used for the COMPAS dataset. This can be easily adapted to other datasets by modifying the task description at the beginning and the examples illustrating the meaning of SHAP scores.

```
1  You are an expert in explainable AI and criminal justice risk assessment.
       Your task is to evaluate the quality of a SHAP explanation that
       describes why a person may be predicted to **recommit a crime**.
2
3  Each explanation is a list of features in the following format:
4  <featureID> <feature_name> : <feature_value> = <feature relevance score>
5
6  Your goal is to determine how **reasonable and high-quality** the
       explanation is, based on the SHAP scores and your domain knowledge.
7
8  ### Understanding SHAP scores:
9  - A positive SHAP score (> 0) means the feature increases the risk of
       recidivism, contributing to a higher predicted risk.
10 - A negative SHAP score (< 0) means the feature decreases the risk,
       contributing to a lower predicted risk.
11 - A SHAP score of 0 means the feature has no impact on the prediction.
12 - The magnitude of the SHAP score reflects the strength of the feature's
       influence on the model's decision - larger absolute values imply
       greater impact.
13
14 ### Your task:
15 Assign a **quality score from 1 to 5**:
16 - **5:** Excellent explanation - all important features have appropriate
       SHAP scores, and no suspicious or unjustified values.
```

```
17 - **4:** Good explanation - mostly reasonable, with at most minor issues
       in some features.
18 - **3:** Moderate quality - some questionable or poorly aligned SHAP
       scores, but overall still partially plausible.
19 - **2:** Poor quality - several features have inappropriate or suspicious
       SHAP scores.
20 - **1:** Very low quality - the explanation is clearly flawed, with major
       issues in multiple key features.
21
22 Also, list **the feature IDs** whose relevance scores are **unjustified
       or suspicious**, based on the feature's value and known importance.
23
24 Do not consider the model's prediction. Focus only on whether the
       explanation is plausible and grounded.
25
26 ### Output format:
27 <score><space><comma-separated list of incorrect feature IDs>
28
29 Examples:
30 - Excellent explanation: '5'
31 - Good explanation with minor issues: '4 5'
32 - Low quality with clear issues: '2 1,6'
33 - Very low quality with major issues: '1 2,4,7'
34
35 If there are no suspicious features, leave the second part empty (just
       the score). DO NOT include any additional text or explanations in
       your response.
```

The obtained scores are then converted into labels following the same procedure as in the user study (see Section 4.2): explanations with scores below three are considered low-quality, while the others are deemed high-quality.

### C.3 HYPERPARAMETER SELECTION

We optimize all hyperparameters using a grid search on the validation split $\mathcal{D}_{val}$. Specifically, for ULER we optimize the SVM kernel (linear, polynomial, RBF), the cost of mistakes $C \in \{0.1, 1, 10\}$, the number of augmentations per explanation $k \in \{5, 10, 20\}$ and the noise $\epsilon_0 \in \{0.1, 0.5, 1\}$. For PASTA, we employ the authors' code for the scoring network and optimize the loss hyperparameters $\alpha \in \{0.1, 1, 10\}$, $\beta \in \{0.001, 0.01, 0.1\}$ and $\gamma \in \{0.01, 0.1, 1\}$. For $\text{NovRej}_X$ and $\text{NovRej}_Z$, we optimize the number of neighbors $k_{NN} \in \{1, 5, 10\}$.

### C.4 ROBUSTNESS TO THE CHOICE OF THE EXPLAINER

**In this section, we assess the robustness of our approach to the choice of explanation method. Specifically, we replicate the experimental setup from Section 4.1, but generate all explanations using LIME (Ribeiro et al., 2016) with its default hyperparameters.**

**Fig. 4 shows the percentage of low-quality explanations for the accepted and the rejected set as a function of the rejection rate $\rho_\%$ averaged across the six datasets considered. Even when using LIME, ULER outperforms the competitors across most rejection rates. On average, across all datasets and rejection rates, ULER reduces the percentage of low-quality explanations in the accepted set by $10\%$ compared to the best competitors $\text{NovRej}_X$ and $\text{NovRej}_Z$.**

**Finally, Table 4 reports the average AUROC per dataset. Again, ULER achieves the highest AUROC on all datasets, demonstrating superior ability to distinguish low- from high-quality explanations. ULER consistently outperforms all baselines in all datasets by improving the AUROC by $13\%$ vs $\text{NovRej}_X$ and $\text{NovRej}_Z$, $18\%$ vs FaithRej and PASTARej, and by more than $22\%$ vs RandRej, ComplRej, StabRej and PredAmb.**

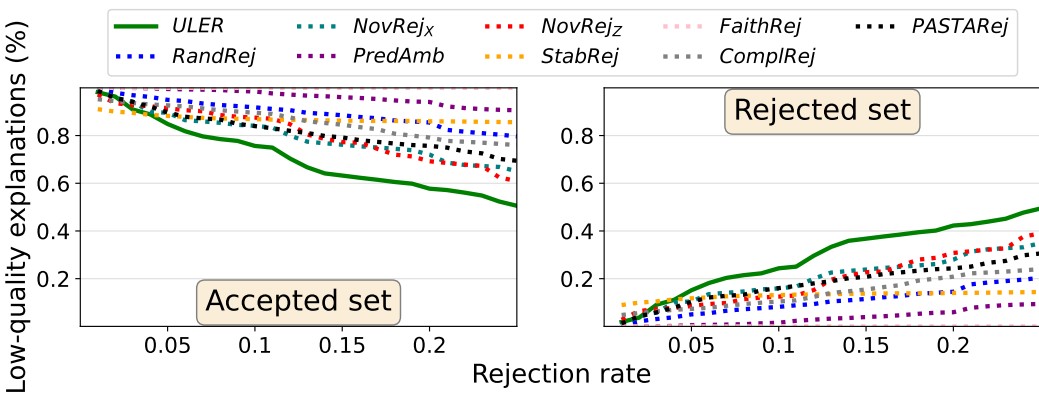

Figure 4: ULER **rejects on average more low-quality explanations than all competitors when** LIME **is used as explainer.** Average percentage of low quality explanations in the accepted and rejected set for all the considered strategies over the 8 datasets for 25 rejection rates $\rho_\%$. ULER outperforms all the competitors for most of the considered rejection rates, demonstrating its robustness to the choice of the explainer.

Table 4: ULER **outperforms the competitors at separating low-quality from high-quality explanations when** LIME **is used as explainer.** Average AUROC for all the rejection strategies over the 8 datasets and its standard deviation. ULER consistently obtains the best results in all datasets, demonstrating its robustness to the choice of the explainer

| | Classification | | | | Regression | | | |
| | compas | creditcard | adult | churn | wine | parkinson | power | bike |
|---|---|---|---|---|---|---|---|---|
| ULER | **0.85 ± 0.16** | **0.57 ± 0.04** | **0.89 ± 0.07** | **0.63 ± 0.04** | **0.58 ± 0.07** | **0.80 ± 0.09** | **0.73 ± 0.04** | **0.62 ± 0.05** |
| RandRej | 0.43 ± 0.25 | 0.53 ± 0.06 | 0.43 ± 0.25 | 0.53 ± 0.06 | 0.52 ± 0.07 | 0.44 ± 0.21 | 0.51 ± 0.10 | 0.51 ± 0.05 |
| PredAmb | 0.41 ± 0.26 | 0.55 ± 0.05 | 0.04 ± 0.05 | 0.45 ± 0.06 | 0.52 ± 0.07 | 0.56 ± 0.18 | 0.52 ± 0.09 | 0.52 ± 0.07 |
| NovRej$_X$ | 0.81 ± 0.12 | 0.51 ± 0.05 | 0.77 ± 0.30 | 0.59 ± 0.04 | 0.51 ± 0.08 | 0.38 ± 0.07 | 0.49 ± 0.06 | 0.52 ± 0.04 |
| StabRej | 0.30 ± 0.17 | 0.52 ± 0.04 | 0.25 ± 0.25 | 0.57 ± 0.07 | 0.46 ± 0.05 | 0.54 ± 0.23 | 0.60 ± 0.06 | 0.45 ± 0.05 |
| FaithRej | 0.55 ± 0.30 | 0.46 ± 0.06 | 0.76 ± 0.05 | 0.58 ± 0.05 | 0.55 ± 0.08 | 0.29 ± 0.18 | 0.55 ± 0.09 | 0.52 ± 0.05 |
| ComplRej | 0.58 ± 0.37 | 0.53 ± 0.04 | 0.30 ± 0.25 | 0.53 ± 0.07 | 0.48 ± 0.06 | 0.30 ± 0.22 | 0.42 ± 0.07 | 0.54 ± 0.04 |
| PASTARej | 0.33 ± 0.34 | **0.57 ± 0.05** | 0.30 ± 0.29 | 0.55 ± 0.08 | 0.50 ± 0.09 | 0.61 ± 0.24 | 0.72 ± 0.04 | 0.60 ± 0.05 |
| NovRej$_Z$ | 0.79 ± 0.24 | **0.57 ± 0.05** | 0.81 ± 0.09 | 0.52 ± 0.08 | 0.54 ± 0.06 | 0.50 ± 0.18 | 0.47 ± 0.07 | 0.47 ± 0.03 |

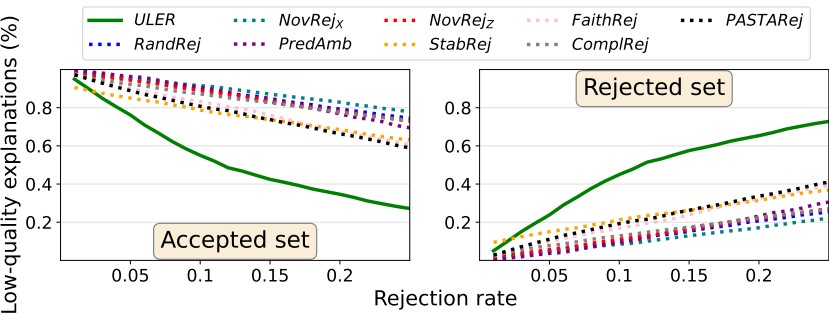

Figure 5: ULER **rejects on average more low-quality explanations than all competitors.** Average percentage of low quality explanations in the accepted and rejected set for all the considered strategies over the 8 datasets for 25 rejection rates $\rho_\%$. For all the considered rejection rates, ULER consistently rejects more low-quality explanations than all competitors.

Table 5: `ULER` **outperforms the competitors at separating low-quality from high-quality explanations.** Average AUROC for all the rejection strategies over the 8 datasets and its standard deviation. `ULER` consistently obtains the best results in all datasets.

| | Classification | | | | Regression | | | |
| --- | --- | --- | --- | --- | --- | --- | --- | --- |
| | compas | creditcard | adult | churn | power | wine | parkinson | bike |
| `ULER` | **0.75 ± 0.04** | **0.87 ± 0.02** | **0.85 ± 0.04** | **0.92 ± 0.01** | **0.90 ± 0.02** | **0.93 ± 0.03** | **0.87 ± 0.01** | **0.78 ± 0.03** |
| `RandRej` | 0.52 ± 0.05 | 0.50 ± 0.02 | 0.53 ± 0.06 | 0.49 ± 0.02 | 0.49 ± 0.02 | 0.51 ± 0.07 | 0.49 ± 0.01 | 0.50 ± 0.07 |
| `NovRej`$_X$ | 0.46 ± 0.04 | 0.58 ± 0.02 | 0.30 ± 0.05 | 0.36 ± 0.02 | 0.46 ± 0.01 | 0.51 ± 0.04 | 0.58 ± 0.02 | 0.54 ± 0.04 |
| `PredAmb` | 0.56 ± 0.03 | 0.46 ± 0.02 | 0.71 ± 0.03 | 0.85 ± 0.01 | 0.49 ± 0.03 | 0.50 ± 0.02 | 0.49 ± 0.02 | 0.51 ± 0.09 |
| `StabRej` | 0.69 ± 0.04 | 0.45 ± 0.02 | 0.53 ± 0.05 | 0.63 ± 0.02 | 0.51 ± 0.02 | 0.76 ± 0.04 | 0.53 ± 0.03 | 0.34 ± 0.06 |
| `FaithRej` | 0.63 ± 0.04 | 0.42 ± 0.02 | 0.71 ± 0.03 | 0.86 ± 0.01 | 0.29 ± 0.02 | 0.74 ± 0.05 | 0.49 ± 0.02 | 0.37 ± 0.04 |
| `ComplRej` | 0.69 ± 0.04 | 0.53 ± 0.05 | 0.45 ± 0.02 | 0.63 ± 0.02 | 0.66 ± 0.02 | 0.62 ± 0.01 | 0.56 ± 0.02 | 0.50 ± 0.05 |
| `PASTARej` | 0.52 ± 0.04 | 0.82 ± 0.03 | 0.66 ± 0.13 | 0.87 ± 0.02 | 0.50 ± 0.03 | 0.55 ± 0.10 | 0.61 ± 0.03 | 0.53 ± 0.10 |
| `NovRej`$_Z$ | 0.46 ± 0.04 | 0.58 ± 0.02 | 0.57 ± 0.04 | 0.52 ± 0.02 | 0.52 ± 0.02 | 0.53 ± 0.05 | 0.57 ± 0.02 | 0.53 ± 0.02 |

## C.5 SIMULATING HUMAN-QUALITY JUDGMENTS WITH A ML ORACLE

**To further validate `ULER`'s effectiveness at rejecting low-quality explanations, we simulate human quality judgments $Y_Z$ and identify features with incorrect relevance scores using a ML oracle $\mathcal{O}$. Specifically, we train a predictor $\mathcal{O}$ and use its explanations $z_{\mathcal{O}}$ as a surrogate for those that an expert would provide. Then we train the proper predictor (that is, $f$) and classify its explanations $z$ as low- or high-quality depending on how much they correlate with the oracle's explanation. In practice, for each classification (resp. regression) task, we train a Random Forest classifier (resp. regressor) to serve as the oracle $\mathcal{O}$ and a linear `SVC` (`SVR`) as the proper predictor. All predictors use the default scikit-learn implementations (Pedregosa et al., 2011). We select predictors with different inductive biases to mirror real-world scenarios where human's predictions may differ from model outputs. Both predictors are evaluated on a disjoint test set consisting of $2000$ instances: the oracle achieves an average balanced accuracy (resp. MSE) of $0.76$ (resp. $0.008$), while the model of $0.69$ (resp. $0.020$).**

**Then, explanations for both the oracle and the predictor are generated on $\mathcal{D}$. An explanation $z$ is labeled as low-quality ($y_z = 0$) if the correlation with the corresponding oracle's explanation $z_{\mathcal{O}}$ falls below a threshold $\tau_z$, and as high-quality ($y_z = 1$) otherwise. We fix $\tau_z = 0.25$ as this ensures datasets with varying amount of low-quality explanations ($1\%$-$48\%$). Additionally, for each explanation $z$, we construct the set of "wrong" relevance scores $\mathcal{W}_z$ by selecting the scores in $z$ that deviate most from the corresponding scores in the oracle explanation $z_{\mathcal{O}}$. Intuitively, if $z$ is low-quality, $\mathcal{W}_z$ should include those entries that account for most of the difference between $z_{\mathcal{O}}$ (which is high-quality by construction) and $z$. To this end, we first compute the difference in relevance $|z_i - z_{\mathcal{O},i}|$ for each $i$, and then include in $\mathcal{W}_z$ the indices $i$'s with the highest difference and that cumulatively account for $u_\%$ of the $L_1$ distance between $z_{\mathcal{O}}$ and $z$. We set $u_\%$ to $0.75$ in the experiments. Since we had sufficient data, we could afford to use non-overlapping sets to train the rejector and the predictor, although doing so is not strictly necessary.**

**Fig. 5 shows the percentage of low-quality explanations for the accepted and the rejected set as a function of the rejection rate $\rho_\%$ averaged over the eight considered datasets. On average, `ULER` reduces the number of low-quality explanations in the accepted set by approximately $24\%$ vs `PASTARej` and `StabRej`, $26\%$ vs `FaithRej`, $32\%$ vs `ComplRej`, $33\%$ vs `PredAmb` and `NovRej`$_Z$, and $34\%$ vs `RandRej` and `NovRej`$_Z$. Moreover, `ULER` rejects the highest number of low-quality explanations in around $94\%$ of the experiments against all competitors. Finally, all the rejectors based on explanation metrics work better than the standard LtR strategies. This confirms that focusing on the prediction ambiguity or input novelty is not aligned with the objective of the LtX setting. Table 5 reports the average AUROC per dataset. `ULER` performs better at separating low-quality from high-quality explanations for all the considered datasets and obtains an average improvement of $21\%$ and $28\%$ from the two runner-ups, respectively `PASTARej` and `ComplRej`.**

Table 6: `ULER` **outperforms its variants that additionally provide inputs and/or predictions as input to the rejector.** Average AUROC for `ULER` and three variants using different inputs to learn the quality of an explanation over the 8 datasets. `ULER` consistently achieves the highest AUROC across all datasets, showing that explanations alone suffice for the rejector to assess their quality.

| | Classification | | | | Regression | | | |
| | compas | creditcard | adult | churn | wine | parkinson | power | bike |
|---|---|---|---|---|---|---|---|---|
| `ULER` | **0.76 ± 0.02** | **0.56 ± 0.03** | **0.71 ± 0.03** | **0.72 ± 0.05** | **0.80 ± 0.05** | **0.59 ± 0.08** | **0.90 ± 0.02** | **0.78 ± 0.03** |
| `ULER`$_{X,Z}$ | 0.71 ± 0.08 | 0.54 ± 0.03 | 0.63 ± 0.05 | 0.48 ± 0.10 | 0.71 ± 0.05 | 0.56 ± 0.15 | 0.79 ± 0.14 | 0.75 ± 0.06 |
| `ULER`$_{Z,Y}$ | **0.76 ± 0.02** | 0.50 ± 0.06 | 0.69 ± 0.02 | 0.65 ± 0.09 | 0.74 ± 0.07 | **0.59 ± 0.06** | 0.89 ± 0.03 | 0.75 ± 0.06 |
| `ULER`$_{X,Z,Y}$ | 0.74 ± 0.06 | 0.54 ± 0.03 | 0.65 ± 0.03 | 0.52 ± 0.11 | 0.65 ± 0.14 | 0.57 ± 0.07 | 0.88 ± 0.05 | 0.76 ± 0.06 |

Table 7: `ULER` **shows a small but consistent improvement over its variant without augmentation in separating low-quality from high-quality explanations.** Average AUROC for `ULER` and `ULER-NOAUG` across the eight datasets. For comparison, we also report `PASTARej`, the best performing baseline. `ULER` consistently achieves a modest but consistent improvement in AUROC across all datasets, while `ULER-NOAUG` still often outperforms `PASTARej`.

| | Classification | | | | Regression | | | |
| | compas | creditcard | adult | churn | wine | parkinson | power | bike |
|---|---|---|---|---|---|---|---|---|
| `ULER` | **0.76 ± 0.02** | **0.56 ± 0.03** | **0.71 ± 0.03** | **0.75 ± 0.06** | **0.72 ± 0.03** | **0.59 ± 0.08** | **0.90 ± 0.02** | **0.78 ± 0.03** |
| `ULER-NOAUG` | 0.70 ± 0.04 | 0.51 ± 0.05 | 0.68 ± 0.02 | 0.71 ± 0.06 | 0.71 ± 0.06 | 0.57 ± 0.04 | **0.90 ± 0.04** | 0.68 ± 0.06 |
| `PASTARej` | 0.66 ± 0.14 | 0.50 ± 0.05 | 0.65 ± 0.04 | 0.53 ± 0.07 | 0.64 ± 0.15 | 0.55 ± 0.06 | 0.74 ± 0.20 | 0.68 ± 0.10 |

## C.6 ULER'S INPUT SPACE

To investigate which inputs the rejector needs to assess explanation quality, we consider three variants of `ULER` in which the rejector works in a different input space: `ULER`$_{Z,X}$ uses both the explanation and its corresponding instance, `ULER`$_{Z,Y}$ uses the explanation along with the prediction, and `ULER`$_{Z,X,Y}$ uses the explanation, the instance, and the prediction. For each variant, we augment the explanations (see Section 3.1), and train the rejector on a training set obtained by concatenating each (augmented) explanation with the input, the prediction, or both.

Table 6 reports the average AUROC per dataset for `ULER` and each of the above variants. Interestingly, including the instances as part of the rejector's input tends to decrease the performance due to the limited number of human-judgment labels which makes it difficult for the rejector to learn the relationship between the explanations and the instances. Moreover, even concatenating only the prediction as in `ULER`$_{Z,Y}$ results in a small performance hit (on average 3%), suggesting that explanations alone are often sufficient.

## C.7 ABLATION STUDY - TRAINING THE REJECTOR WITHOUT AUGMENTING THE DATA

In this section, we evaluate whether augmentation improves the rejector's performance, and thus whether collecting per-feature feedback is beneficial. To this end, we compare `ULER` with an ablated variant, `ULER-NOAUG`, which does not leverage the feedback-aware augmentation strategy (*i.e.*, does not exploit the per-feature feedback). Specifically, `ULER-NOAUG` trains the rejector as described in Section 3.1, but uses $\mathcal{D}$ instead of the augmented data $\mathcal{D}_{aug}$.

Table 7 reports the average AUROC per dataset for both `ULER` and `ULER-NOAUG`, assessing their performance in distinguishing low-quality from high-quality explanations. For comparison, we also report `PASTARej`, the best-performing baseline in Q1. `ULER` consistently outperforms its ablated variant across all considered datasets. While the performance gain in performance is quite small ($\approx$ 3%), it in consistent: `ULER` always outperforms the variant without augmentation across all datasets. We argue that this improvement is still worth it given the minimal additional cost to obtain the feature-level feedback. Once user-provided quality judgments are collected, obtaining per-feature feedback is inexpensive because users are already focused on identifying features with wrong scores to assess explanation quality. In cases where per-feature feedback is not available, one could skip the augmentation step and simply use `ULER-NOAUG`, which still consistently outperforms `PASTARej` across most datasets, improving the AUROC by approximately 7%.

Table 8: ULER **is not strongly correlated with existing machine-side metrics.** Average Spearman correlation coefficient ($\pm$ std) between ULER and each machine-side metric on the user study data.

|  | faithfulness | stability | complexity |
|---|---|---|---|
| user study | -0.43 $\pm$ 0.17 | -0.22 $\pm$ 0.10 | -0.05 $\pm$ 0.12 |

Table 9: ULER **predicts the human-judgments better than all competitors.** Average AUROC and its standard deviation for all the rejection strategies on the user study data.

| rejector | AUROC ($\pm$ std) |
|---|---|
| ULER | **0.64 $\pm$ 0.05** |
| RandRej | 0.47 $\pm$ 0.08 |
| PredAmb | 0.46 $\pm$ 0.07 |
| NovRej$_X$ | 0.39 $\pm$ 0.07 |
| StabRej | 0.43 $\pm$ 0.10 |
| FaithRej | 0.44 $\pm$ 0.07 |
| ComplRej | 0.45 $\pm$ 0.03 |
| NovRej$_Z$ | 0.49 $\pm$ 0.06 |

### C.8 Correlation analysis with machine-side metrics for user-study data

We repeat the same setup as in Q1, but compute the Spearman coefficient on the user study data. Again, we do not observe strong correlations, confirming that ULER captures information that is different from existing machine-side metrics.

### C.9 Q3: Comparison with the other competitors

Additionally, we replicate the same experiments described in Section 4.2 including all competitors in Section 4 to further validate that standard LtR strategies and machine-side metrics cannot reliably reflect user judgments.

Table 9 reports the average AUROC for ULER and the other seven competitors (results for PASTARej are reported in the main paper), measuring their ability to distinguish between high-quality and low-quality explanations. ULER outperforms all competitors, achieving at least an 15% improvement in AUROC and demonstrating more consistent performance, as indicated by the lower standard deviation. Moreover, we observe that the explanation-aware strategies perform similarly to the random rejector, thus confirming that existing machine-side metrics do not capture human judgments.

Additionally, we found that human annotators identified, on average, 1.8 features with incorrect relevance scores in low-quality explanations, compared to only 0.7 features in high-quality ones. This supports our intuition that low-quality explanations are perceived by users as containing more wrong relevance scores.

## D  User study

### D.1  Data

For this user study, we used the publicly available StatsBomb 360 event stream data (Statsbomb, 2023). This contextualized event stream data is extracted from broadcast video and contains event stream data, and snapshots of player positioning at the moment of each event. The event stream data describes semantic information about the on-the-ball actions, such as which actions are performed, their start and end location, the outcome of the action, which players performed them, and the time in the match they were performed at.

## D.2 OBTAINING THE EXPLANATIONS

To obtain the explanations, we begin by preprocessing the data (Statsbomb, 2023) to obtain the features needed to train the classifier. From each shot snapshot, we extract the following features: (i) the distance from the ball to the center of the goal, (ii) the angle between the ball and the goalposts, (iii) the distance of the goalkeeper from the goal line, (iv) the distance of the goalkeeper from the midline (*i.e.*, the line that passes through the center of the field and the middle of the goals), and (v) the distance to the closest defender (excluding the goalkeeper). We select only these features for two main reasons: they are easily interpretable from the snapshot (see Fig. 3), and their meanings are non-overlapping, which makes it easier for annotators to disentangle their individual contributions as we found empirically that working with strongly correlated features can complicate human assessment. Using these features, we train an XGBoost ensemble (Chen and Guestrin, 2016) consisting of 50 trees with a maximum depth of 3, as it is standard practice in soccer analytics (Robberechts et al., 2020). The model is trained on shots from the 2015–2016 season across four major top-tier leagues (Germany, Spain, England, and France). We evaluate the classifier on a held-out test set of 1,050 shots from the Italian top division in the same season. The primary goal of xG is to produce well-calibrated probability estimates because they are used for decision making (e.g., evaluating players and giving advice about when to shot), which we assess by reporting the Brier score. Additionally, goals should receive an higher scoring probability than non-goals, which we capture by using AUROC. The model achieves a Brier score of 0.067 and an AUROC of 0.81.

We then use the test set to generate the explanations. As for the benchmark datasets, explanations are generated using Kernel`SHAP` (Lundberg and Lee, 2017) with 100 samples and the training set used as background.

## D.3 HUMAN ANNOTATION PROCESS

Participants were recruited using Prolific, a crowd sourcing platform. We applied Prolific's filters to ensure that participants possessed sufficient soccer expertise. Specifically, we applied filters to recruit subjects that (*i*) live in countries where soccer is widespread (UK, Germany, France, Spain, Belgium, Italy, Netherlands, or Portugal), and (*ii*) actively watch and play soccer. All participants were compensated with £3 for an expected completion time of 25 minutes, as estimated from the pilot studies.

After conducting pilot studies to ensure that the task was clear and comprehensible and to verify intra-annotator consistency, we launched the main user study. Participants were first requested to give their consent to participate. Then, they were provided with a link to an external Google Doc containing task instructions, which they could consult at any time during the session. The document provides general introduction for the task setting and objective, the description and illustration of the predictor's features, and 3 exemplary snapshots.

After the task introduction, participants completed three warm-up trials to familiarize themselves with the interface and the task; this was followed by the real annotation session comprising of 30 trials. In each trial, participants were asked three questions: two 5-point Likert-scale questions to separately assess the quality of the prediction and explanation, and one multiple choice question to identify the features with a wrong relevance score. We used two separate questions, presented in distinct sections of the form, to disentangle participants' agreement with the prediction from their perception of the explanation's quality and to minimize spurious correlations between their responses. 5-point Likert scales have been chosen as they provide satisfactory reliability and validity (Taherdoost, 2019). Specifically, in the first question, participants were shown an image containing only the shot snapshot along with the predicted probability of scoring (see Fig. 6) and asked to assess their agreement with the prediction - *"The AI thinks that the probability that the shooter will score is 1%, which is much lower than the average (10%). To what extent do you agree with the AI's prediction?"*, where the comparison *much lower* was dynamically adapted based on the predicted probability. For the second and third questions, participants were shown a different image containing the shot snapshot, the prediction, and the explanation (see Fig. 7). To facilitate interpretation, features relevance are visualized as independent arrows: blue indicates a positive impact on the prediction, while red indicates a negative impact. The second question - *"To what extent is the AI's explanation consistent with how you would explain the predicted probability of scoring?"* - was used to collect the perceived explanation quality. While the third question - *"Which features are being*

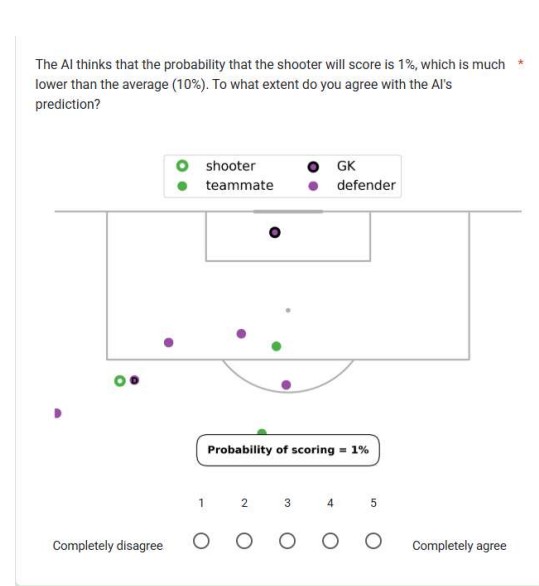

Figure 6: Example of the first image of each trial

*used incorrectly, if any?"* - is used to obtain the feature-level feedback about the features with an incorrect relevance score in the prediction. To ensure high-quality annotations, we included an attention check requiring specific answers for a trial. This allowed us to detect and discard inattentive or randomly answering participants.

### D.4 ANNOTATIONS PREPROCESSING

To ensure high-quality annotations, we applied several filtering steps. First, we excluded participants who failed more than one attention check question, as well as those who consistently provided the same score for every explanation (typically a score of 3), since this means they were not able (or did not bother) to discriminate between explanations. We also removed two participants who did not flag any relevance score as incorrect. Additionally, given the subjective nature of the task (for instance, we saw that showed very low annotator agreement, *e.g.*, 1 vs 5) we removed explanations for which the standard deviation of the explanation quality scores exceeded $1.25$. This step helped ensuring that our dataset contains only explanations where annotators' opinions are reasonably consistent. After applying these filters, $718$ explanations remained for our experiments.

## E   LLM USAGE

LLMs were used to polish the writing, to rephrase sentences, and to debug the code. Our manuscript and our code was first human-generated, and then possibly enhanced by LLMs.

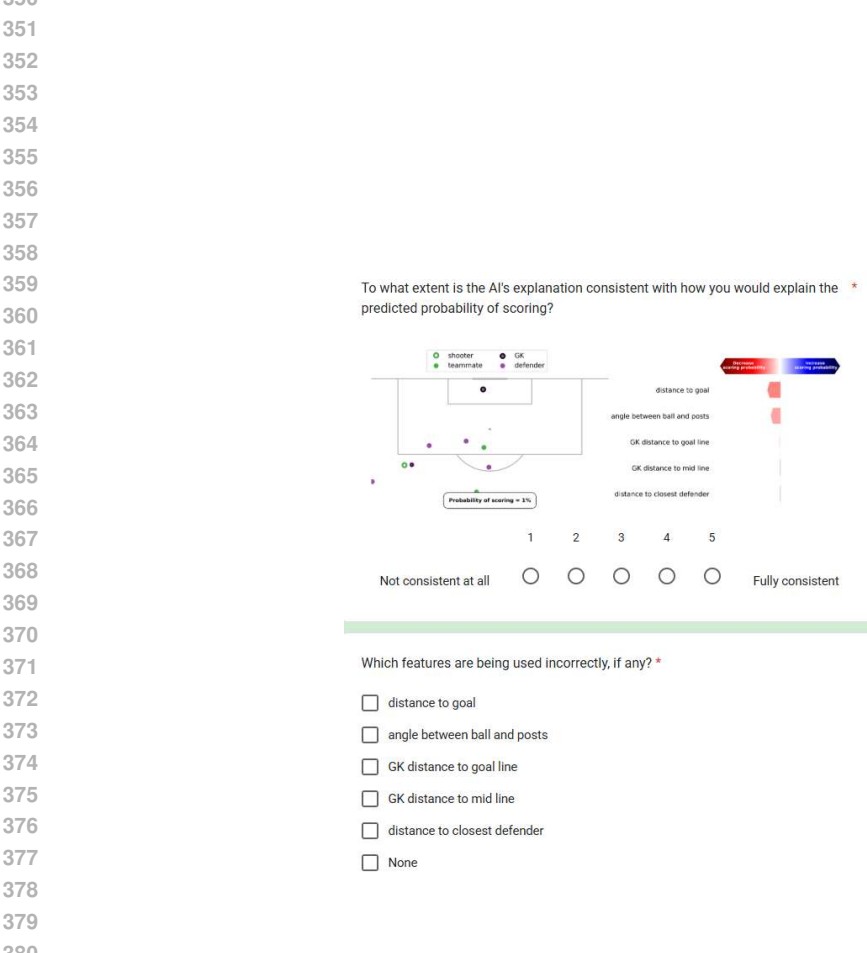

Figure 7: Example of the second image of each trial

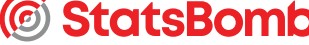

