# OpenReview forum: "Learning to Reject Low-Quality Explanations via User Feedback"
_ICLR.cc/2026/Conference — Submitted to ICLR 2026_

### Official Review · Reviewer_zyBV · 2025-10-24

**Soundness:** 2
**Presentation:** 2
**Contribution:** 1
**Rating:** 2
**Confidence:** 4

**Summary:**

This authors introduce Learning to Reject Low-Quality Explanations (LtX), extending the traditional Learning to Reject framework to consider explanation quality as a criterion for deferring predictions. The proposed method, ULER (User-centric Low-quality Explanation Rejector), trains a rejector to predict when explanations are rated as low-quality by users. ULER leverages a small set of human-labeled data and optional per-feature feedback through a data augmentation scheme. Experiments on eight tabular benchmarks with simulated human judgments and a new human-annotated soccer dataset show that ULER outperforms standard LtR methods and PASTARej in filtering out low-quality explanations.

**Strengths:**

- The authors address a crucial yet under-explored dimension of trustworthy AI - the quality of explanations as a basis for deferring predictions to human experts. This is an important step toward integrating XAI into selective prediction frameworks.

- ULER’s formulation is modular, combining human feedback, lightweight data augmentation, and a threshold-based rejector. The design is compatible with existing explainers like SHAP or LIME and does not depend on model architecture.

- The experiments are extensive, spanning multiple tabular benchmarks and a new human-annotated dataset, and the authors detail datasets, hyperparameters, and reproducibility plans.

- The authors plan to release a new dataset of human-rated explanations, which will be valuable to the community and help standardize evaluation of human-aligned XAI metrics.

**Weaknesses:**

- The paper treats explanation quality as equivalent to human agreement, but it does not clearly define the conceptual dimensions this label captures or explain why they are important.

- Previous work has highlighted that explanations can appear plausible to humans without accurately reflecting the model’s reasoning, motivating metrics that assess faithfulness rather than plausibility [1,2,3]. This paper, instead, emphasizes incorporating human judgment into the Learning to Reject framework over machine-side metrics. The motivation and rationale for this alternative perspective should be more clearly articulated.

- From the experiments, it appears that the paper assumes human experts understand the ground-truth data-generating process and can reliably identify important features. Making this assumption explicit would help clarify why human judgment is prioritized over machine-side metrics, and better define the range of tasks that ULER can be applied on (tasks where human know the ground-truth data-generating process)

- ULER and PASTA both aim to approximate human judgments of explanation quality. While the paper lists three main differences - PASTA is designed for image data with an embedding network, lacks a rejection mechanism, and seeks a dataset-agnostic metric - the experimental setup for PASTARej removes the embedding network, fits its scoring network to predict human judgment, and uses it for rejection. The paper could more clearly clarify the conceptual distinctions between ULER and PASTARej.

- The benchmark experiments rely on simulated human judgments generated by an LLM. Since ULER (and PASTARej) is trained on these labels while other baselines are not, the reported performance advantage may be partially influenced by this setup.

- Converting 5-point Likert responses into binary labels reduces annotation noise but removes information about gradations in perceived quality. This simplification may limit the rejector’s ability to handle borderline or ambiguous cases that could be practically relevant.

[1] Chowdhury, Townim, et al. "Looking in the Mirror: A Faithful Counterfactual Explanation Method for Interpreting Deep Image Classification Models." Proceedings of the IEEE/CVF International Conference on Computer Vision. 2025.
[2] Agarwal, Chirag, Sree Harsha Tanneru, and Himabindu Lakkaraju. "Faithfulness vs. plausibility: On the (un) reliability of explanations from large language models." arXiv preprint arXiv:2402.04614 (2024).
[3] Lu, Xiaolei, and Jianghong Ma. "Does faithfulness conflict with plausibility? an empirical study in explainable ai across NLP tasks." arXiv preprint arXiv:2404.00140 (2024).

**Questions:**

- Could you formalize what “user-perceived explanation quality” entails beyond trust? Are there specific criteria that distinguish a high-quality explanation from a low-quality one?

- Could you clarify why human judgment is preferred over machine-side metrics for assessing explanation quality, especially considering prior work suggesting that explanations can appear plausible to humans without faithfully reflecting the model’s reasoning?

- Does your framework assume that human experts understand the ground-truth data-generating process and can reliably identify important features? If so, how might this assumption limit the types of tasks where your framework is applicable? If not, how should readers interpret the reliability of human judgment in this context?

- Could you clarify the distinctions between ULER and PASTARej, particularly in the context of your experiments?

- The paper relies on simulated judgments from an LLM. Were these validated against real human ratings, and if not, how should readers interpret the benchmark results?

- In lines 212–216, it is stated that “ULER tends to yield marginal improvements in predictive performance for the accepted inputs by rejecting explanations of incorrect predictions.” Which experimental results directly support this claim?

- How adaptable is ULER to non-feature-based explanations (e.g., textual rationales or counterfactual examples)? Is the framework specific to feature attributions?

---

> ### Author Response · Authors · 2025-11-24
>
> Dear Reviewer zyBV,
>
> We appreciate your valuable and insightful feedback that will improve the paper! We are glad that you consider our paper an important step toward the integration of XAI into Learning to Reject frameworks, our experiments *extensive* and you recognize the value for the community of the new dataset we will release. Here are the answers to the remaining concerns:
>
> > **Explanation quality equivalent to human agreement.** Could you please elaborate on this point?
>
> > **Human judgment over machine-side metrics.** While many machine-side metrics for explanation quality exist in the literature, it has been shown that they often fail to predict human judgments[1,2], a gap that is still overlooked. This is why our approach focuses on learning a rejector that assesses explanation quality from the user’s perspective, allowing low-quality explanations to be filtered out. At the same time, to avoid presenting, e.g., unfaithful explanations to users that may appear persuasive, one can always combine ULER with a faithfulness-based rejector (e.g., FaithRej).
>
> > **Human judgments provided by domain experts.** We assume that human‐quality judgments and per‐feature feedback are provided by domain experts (lines 176–177). This does not require experts to understand the whole ground-truth data-generating process; rather, they must be able to assess whether an explanation is plausible and aligned with their domain knowledge. Our framework is therefore applicable to high-stakes domains where expert validation is increasingly becoming a legal requirement (lines 46-49).
>
> > **Conversion of 5-point Likert scales to binary labels.** We agree that converting 5-point Likert responses into binary labels removes information about gradations in perceived quality. Learning directly from the full Likert scores could be an interesting direction for future work.
>
> > **Formalization of “user-perceived explanation quality”.** We clarified in the text that, in our framework, user-perceived explanation quality reflects two complementary dimensions: plausibility, meaning that the relevance scores should align with the user’s domain knowledge, and interpretability, meaning that the explanation should be understandable by the user.
>
> > **ULER’s marginal improvement in predictive performance for the accepted inputs.** We thank the reviewer for the comment. Our statement was meant to convey that ULER may reject some incorrect predictions by filtering out explanations deemed low-quality. To effectively improve the predictive performance, ULER should be combined with state-of-the-art learning-to-reject strategies, which are specifically designed for this purpose. We clarified this in section 3.2 Benefits and Limitations.
>
> > **ULER for non-feature-based explanations.** We thank the reviewer for the interesting question. ULER’s augmentation strategy is indeed tailored to feature-based explanations, as it perturbs relevance scores. For non-feature-based explanations, this specific mechanism would not directly apply. However, the core idea of ULER, i.e., learning a rejector from human judgments of explanation quality, is more general and could still be explored without the augmentation step. Investigating how to adapt ULER to non-feature-based explanations is an interesting direction for future work.
>
> [1] Colin, Julien, et al. "What i cannot predict, i do not understand: A human-centered evaluation framework for explainability methods." Advances in neural information processing systems (2022): 2832-2845.
>
> [2] Kazmierczak, Rémi, et al. "Benchmarking xai explanations with human-aligned evaluations." arXiv preprint arXiv:2411.02470(2024).

---

### Official Review · Reviewer_fLkm · 2025-10-30

**Soundness:** 1
**Presentation:** 2
**Contribution:** 2
**Rating:** 2
**Confidence:** 4

**Summary:**

The paper introduces a framework for identifying and rejecting low-quality training samples to improve the performance of ML models. The authors formulate this as a learning-to-reject problem, training a rejection function alongside the primary predictive model to detect data samples that may be noisy, mislabeled, or otherwise detrimental to model generalization. The proposed approach uses meta-learning to optimize the rejection mechanism with respect to a held-out validation objective. Experiments across multiple tabular datasets demonstrate improved generalization and robustness compared to baseline data filtering or robust training methods.

**Strengths:**

The authors run extensive experiments across eight diverse tabular benchmarks and a real-world soccer analytics user study with 5,250 human annotations.

**Weaknesses:**

1. The paper evaluates all competitors on eight benchmark datasets using simulated human judgments generated by Llama-3.1-8B-Instruct. While using LLMs as judges is an emerging and sometimes practical trend, the choice of such a small model (8B) raises **serious reliability concerns.** The core issue is not the general idea of using an LLM judge, but rather the limited evaluative and reasoning capacity of this particular model size. Current literature suggests that larger models (e.g., GPT-4, Claude 3, or at least Llama-3.1-70B) align more closely with human judgments, whereas 8B-class models exhibit inconsistent performance, particularly on domain-specific tasks. Results derived from Llama-3.1-8B-Instruct (Q1 and Q2) should be treated as exploratory or diagnostic, rather than definitive empirical evidence.
2. Evaluating the quality of explanations in XAI has been widely studied, with numerous metrics and criteria already established. The concept of learning to reject or penalizing low-quality explanations is not novel. It has antecedents in selective prediction, abstention learning, and robust loss design. Modern approaches such as Direct Preference Optimization (DPO) offer a more principled and interpretable framework for learning from preferences or penalties, especially in the post-LLM era. Although I am not a big fan of LLM, the idea behind the current work and the experimental setting should not be targeted at ICLR.
3. Presentation issue: most figures could be improved for professionalism, clarity and readability. Moreover, the mathematical exposition of the rejection function lacks necessary precision.

**Questions:**

See above.

---

> ### Author Response · Authors · 2025-11-24
>
> Dear Reviewer fLkm,
>
> We appreciate your valuable and insightful feedback that will improve the paper! We are glad that you appreciated our extensive empirical evaluation and we will do our best to improve the clarity and the readability of the figures. Here are the answers to the remaining concern:
>
> > **Paper should not be targeted at ICLR.** Could you elaborate on this point please? We believe that works on explainability, human interaction and learning to reject are a perfect fit for the conference.

---

> > ### Comment · Reviewer_fLkm · 2025-11-25
> >
> > Thank you for the clarifications. I appreciate the additional context regarding resource constraints and the ML-oracle experiment. However, my initial concern remains. While I acknowledge the computational or financial constraints in practice, my concern is methodological rather than logistical. Your framing positions ULER as a method for rejecting low-quality explanations from the user perspective, but the majority of the empirical evidence still depends on judgments produced by a small model.
> >
> > Regarding the paper positioning and targeting, I agree that the overall topic fits the conference well, but the current submission does not meet the typical expectations of ICLR, in terms of the overall presentation and empirical rigor.

---

### Official Review · Reviewer_4xMk · 2025-10-31

**Soundness:** 2
**Presentation:** 2
**Contribution:** 2
**Rating:** 4
**Confidence:** 3

**Summary:**

This paper introduces a new problem setting called "Learning to Reject Low-Quality Explanations" (LtX), which argues that machine learning models in high-stakes domains should not only provide predictions but also have the ability to withhold predictions when their accompanying explanations are of low quality from a user's perspective. To address this, the authors propose a novel framework called "User-centric Low-quality Explanation Rejector" (ULER). ULER trains a rejector model on a small set of human-annotated explanations to learn a policy for identifying and rejecting unsatisfactory explanations. A key component of ULER is a data augmentation strategy that leverages per-feature feedback from users to create a more robust training set for the rejector. The authors conduct a comprehensive empirical evaluation of ULER on eight benchmark datasets with simulated human feedback and a new human-annotated dataset from a real-world soccer analytics task. The results demonstrate that ULER outperforms existing "Learning to Reject" (LtR) strategies and various explanation quality metrics in identifying and rejecting low-quality explanations, and is better at mimicking human judgments.

**Strengths:**

- The paper introduces the novel and well-motivated problem of "Learning to Reject Low-Quality Explanations" (LtX), which addresses a critical gap in the existing "Learning to Reject" (LtR) literature that has traditionally focused only on prediction quality while ignoring the quality of the accompanying explanations.

- The proposed ULER framework is user-centric, directly learning from human feedback to align the rejection mechanism with human judgments of explanation quality. The use of both high-level quality ratings and more granular "per-feature relevance judgments" to inform the rejector is a sensible approach. The data augmentation strategy, which perturbs "the features with correct relevance scores" for low-quality explanations, is a creative method to expand the training data from a small set of annotations.

- The empirical evaluation is extensive, covering eight benchmark datasets with simulated human judgments and a user study with a newly collected human-annotated dataset. The authors' commitment to creating and releasing "the first larger-scale (1050 examples, 5 annotations each) data set of human-annotated explanations" is a valuable contribution to the research community that will facilitate future work in this area. The user study itself is thoughtfully designed, with considerations for participant expertise, clear instructions, and filtering criteria to ensure data quality (Section 4.2).

**Weaknesses:**

- The framing of ULER as "explainer-agnostic" is a significant overstatement. The methodology is fundamentally tied to feature attribution methods that produce a "relevance score zi ∈ R to each input feature xi" . The entire ULER framework, including the per-feature feedback ("indicate as Wz (resp. Cz) the indices of the features whose relevance the user deems wrong (resp. correct)") and the data augmentation strategy, is designed around manipulating these feature importance vectors. It is unclear how ULER would be applied to other important classes of explanations, such as counterfactual explanations, example-based explanations, or concept-based explanations, which do not naturally produce per-feature relevance scores. This limitation significantly narrows the practical applicability of ULER beyond the realm of feature attribution methods.

- The use of a LLM (Llama-3.1-8B-Instruct) to "simulate human quality judgments" for the benchmark experiments is a major methodological weakness. While the authors followed an existing approach, this simulation introduces a significant potential for confounding variables. An LLM's "understanding" of feature importance and its alignment with human intuition is not guaranteed and is heavily dependent on the prompting strategy. The paper presents the results from these simulated experiments as strong evidence of ULER's effectiveness, but it is plausible that ULER is simply better at modeling the specific artifacts and biases of the LLM used for data generation rather than genuine human reasoning. The paper lacks a critical discussion of the limitations of this simulation and how it might affect the validity and generalizability of the findings from the benchmark datasets. (There also seems to be some relevant work in this direction: https://openreview.net/pdf?id=MOtZlKkvdz)

- The experimental design has a potential flaw in its choice to use only KernelSHAP for generating explanations. While KernelSHAP is a popular method, different explanation techniques have distinct properties and failure modes. The paper's conclusions about the ineffectiveness of machine-side metrics (e.g., stability, faithfulness) might be specific to KernelSHAP. It is possible that for other explainers, these metrics could be more effective at identifying low-quality explanations. By not evaluating ULER with a more diverse set of explainers (e.g., LIME, Integrated Gradients), the paper misses an opportunity to demonstrate the true robustness and generalizability of the proposed approach. The claim that ULER can "assess the perceived quality of attributions irrespectively of how these are computed" is not sufficiently supported by the experiments.

- The scalability and practicality of the proposed annotation process raise concerns. The authors state that ULER "uses modest amounts of human annotations", but the user study still required collecting 5250 annotations for a single dataset with only a few features. In many real-world applications, models can have hundreds or thousands of features, which would make the per-feature feedback process ("select individual features they believed were misused in the explanation") prohibitively time-consuming and cognitively demanding for human annotators. The paper does not adequately address how the ULER framework would scale to such high-dimensional settings, which could be a significant barrier to its adoption in practice.

**Questions:**

None.

---

> ### Author Response · Authors · 2025-11-24
>
> Dear Reviewer 4xMk,
>
> We appreciate your valuable and insightful feedback that will improve the paper! We are glad that you found our problem statement *novel and well-motivated*, our approach *sensible* and our experimental evaluation *extensive*. Here are the answers to the remaining concerns:
>
> > **Evaluating ULER with a diverse set of explainers.** We thank the reviewer for the thoughtful feedback. We added a new experiment in Appendix C.4 to show that the results remain consistent for the simulated setting also when LIME is used as the explainer. As expected, also when using LIME as the explainer, ULER is always better at discriminating low-quality from high-quality explanations and improves the AUROC by 12% wrt the best competitor, NovRej_Z.
>
> > **High-number of annotations required for the user study.**  We clarify that ULER was trained using only 574 human-quality judgments, as reported in Section 4 (Setup) and Section D.5. Accordingly, we consider the annotation requirements to be modest and feasible for real-world deployment. Although ULER does not require a large number of labels for training, we collected a larger set of annotated explanations to enable a robust and reliable evaluation of its performance.
>
> > **ULER is explainer-agnostic.** Thank you for pointing this out. We clarified the point in the paper.
>
> > **It is plausible that ULER is simply better at modeling the specific artifacts and biases of the LLM.** We politely disagree: ULER outperforms the competitors even in our larger-scale user study, where responses are not simulated. Concerning the fact that LLMs can exhibit artifacts and biases, we remark that: i) we used them as they are the best surrogate of human judgments we can afford, ii) humans, even experts, are also affected by judgment biases, as documented by a wealth of work in behavioral psychology [1]. The fact that ULER works for LLMs (and for SVMs and Random Forest oracles, see our general comment above) suggests that it can work irrespective of who/what provides the explanation quality judgments.
>
> > **ULER framework in high-dimensional settings.** Asking for feature-level feedback is a standard practice in other high-dimensional domains[2,3]. In such domains, obtaining per-feature feedback can be more cognitively affordable by displaying only a limited number of top-ranked features (i.e., those with the highest relevance scores) which users can reasonably assess. In practice, users are expected to flag either (i) features among the presented one whose scores they believe are incorrect, or (ii) additional features not shown but which they would expect to have significant importance. We added this discussion in the Methodology section.
>
> [1] Bertrand, A., Belloum, R., Eagan, J. R., & Maxwell, W. (2022, July). How cognitive biases affect XAI-assisted decision-making: A systematic review. In Proceedings of the 2022 AAAI/ACM Conference on AI, Ethics, and Society (pp. 78-91).
>
> [2] Teso, Stefano, et al. "Leveraging explanations in interactive machine learning: An overview." Frontiers in Artificial Intelligence 6 (2023): 1066049.
>
> [3] Teso, Stefano, and Kristian Kersting. "Explanatory interactive machine learning." Proceedings of the 2019 AAAI/ACM Conference on AI, Ethics, and Society. 2019.

---

### Official Review · Reviewer_JGDU · 2025-11-01

**Soundness:** 3
**Presentation:** 3
**Contribution:** 2
**Rating:** 6
**Confidence:** 3

**Summary:**

This work focuses on developing a framework such that predictor models can learn to reject when explanations are low-quality.
* The authors propose ULER (User-centric Low-quality Explanation Rejector), a rejector model trained on human annotated data and per-feature relevance judgements.
* ULER outperforms other strategies on eight benchmarks and a new human-annotated soccer dataset.
* The authors also release the dataset of human-annotated explanations in the soccer domain.

**Strengths:**

* The method addresses an important alignment gap: explanation quality is important for high-stakes applications but is typically not considered in rejection strategies.
* The authors provide comprehensive results across many standard benchmark datasets, including a new dataset they created and released.
* The methodology is well presented and straightforward.

**Weaknesses:**

* As acknowledged in the paper, the work focuses on tabular data and the approach requires human annotations, which limits how generalizable it is.
* Many of the baselines (besides PASTA) are not optimized/designed for rejecting low-quality explanations, so the comparison on that metric feels a bit biased. Also, the method seems to be quite similar to PASTA except for the rejection framing. The paper could be improved by including some of the details on the differences to PASTA from Appendix B.2 in the main body.
* The method seems to be reliant on having a clear ground truth for the per-feature annotations. How would it perform if there is annotator disagreement or if the features are highly correlated?

**Questions:**

How did you assess the quality of the simulated human judgements from Llama-3.1-8B?

---

> ### Author Response · Authors · 2025-11-24
>
> Dear Reviewer JGDU,
>
> We appreciate your valuable and insightful feedback that will improve the paper! We are glad that you found the paper *well-written and straightforward* and the experimental set-up is *comprehensive*. Here are the answers to the remaining concerns:
>
> > **Focus on tabular data.** The focus on tabular data is an explicit design choice of our approach. However, we agree that extending ULER to other data modalities is an interesting direction for future work.
>
> > **Human annotations required.** We agree that this is a limitation of our approach and we explicitly acknowledge it in Section 3.2 (lines 223–226). However, we believe this limitation is necessary: assessing explanation quality from the user perspective fundamentally requires human judgments, as shown in Section 4.2.
>
> > **Annotators’ disagreement.** We expect per-feature labels to be provided by a single domain expert, so annotator disagreement should not arise in our setting. Nevertheless, as shown in Section C.5, ULER remains effective even when per-feature labels are unreliable or unavailable: in such cases, one can simply skip the augmentation step and train the rejector directly on human-quality judgments. This variant still outperforms all baselines, improving AUROC by at least 7%.

---

### Author Response · Authors · 2025-11-24

We thank all reviewers for their thoughtful feedback and constructive suggestions. We have uploaded a revised version of the manuscript that incorporates the additional experiments you recommended and clarifies several points raised in the reviews. All modified text is shown in violet. Below you find the answers to the shared concerns:

> **Differences between ULER and PASTARej.** We emphasize that PASTA was designed as a universal metric to assess explanation quality from a human perspective, whereas our goal is to detect and reject low-quality explanations within a specific dataset. To this end, we introduce a feedback-aware augmentation strategy that enables the rejector to learn to distinguish between high- and low-quality explanations. We have clarified this distinction in the main body of the paper (Section 4 - Competitors).

> **Biased experimental comparison.** Our core claim is indeed that machine-side metrics alone are insufficient to detect explanations that are low-quality from the user’s perspective (lines 75-77), which is precisely what our experiments demonstrate. Assessing explanation quality from the user perspective requires collecting human judgments, which are then used to train ULER to reject low-quality explanations, allowing us to go beyond what machine-side metrics can capture.

> **Llama-3.1-8B to simulate human-jugments.** First, we highlight that our main empirical result is the user study experiment, where we demonstrate on user-annotated data that ULER outperforms all competitors at rejecting low-quality explanations. We include experiments on benchmark datasets primarily to show that the results remain consistent also in this setting. Second, obtaining such a large number of simulated human-judgment labels (8 datasets × 2000 instances = 16,000 labels) from models like GPT-4 or Claude 3 would be prohibitively expensive. On the other hand, Llama-3.1-70B requires substantial computational resources due to its high memory demands, which unfortunately exceed the capacity of our servers.
To further validate our findings, we additionally include in Appendix C.5 an experiment where we employ a ML oracle to simulate the domain expert and obtain the simulated human-judgments. In this setting, an explanation is deemed low-quality if the machine explanation differs significantly from the oracle’s explanation. Again, ULER consistently outperforms all baselines by rejecting a higher number of low-quality explanations and by better discriminating low-quality from high-quality explanations. Together, these results strengthen the claim that ULER is not simply modeling artifacts of a particular simulation (LLM or ML oracle), but is robust across different sources of simulated judgments. Across both human-annotated data and the two distinct simulation paradigms (LLM-based and ML-oracle-based), ULER consistently rejects more low-quality explanations than existing LtR strategies and explanation-metric-based strategies.

> **Validation of the simulated human-judgment labels.** To assess the quality of the simulated human judgments produced by Llama-3.1-8B, we manually inspected a representative sample of the labels to ensure that the judgments were coherent and aligned with human expectations. While this provides a sanity check, fully validating the simulated human judgments would require conducting a user study for each dataset, which is prohibitively expensive and challenging, especially in domains that demand expert knowledge.

---

### Meta-Review · Area_Chair_2Wmd · 2026-01-08

**Summary:**

**Summary** This paper extends traditional learning-to-reject methods to consider explanation quality as a criterion for selective classification. The authors propose a rejector trained on human-annotated explanation ratings and per-feature relevance judgments, combined with a data augmentation strategy to expand limited training data. The approach is evaluated on tabular benchmarks using simulated human judgments and a new human-annotated dataset. Results show the proposed method outperforms standard learning-to-reject baselines at filtering low-quality explanations.

**Review Process** This paper received four reviews with substantive engagement. Two reviewers recommended rejection, one recommended weak reject, and one recommended weak accept. On the one hand, reviewers recognized the paper was polished, well-developed, with an extensive experimental evaluation. On the other hand, reviewers raised concerns about the motivation for explanations based on user preferences and the soundness/validity of certain claims (e.g., using LLM-simulated human judgments). There were a number of complaints in this case that are common for papers in this area that I ignored (e.g., PASTA).

**Recommendation/Rationale**  Having read the reviews, the rebuttal, and the paper, I am unfortunately recommending rejection. My decision is mostly based on the strong views in the initial reviews. Even as the rebuttal did address some of these concerns, some of the most important ones remain unresolved. I don't see how this would have been overturned in a rebuttal – especially given the lack of a champion among initial reviews and the breadth of complaints. In the interest of highlighting signal over noise, I will state that it is not normal for a paper that is this polished to receive these kinds of scores – which often leads to superfluous feedback from reviewers who spot something is off but can't articulate it well.

On my end, I personally find the questions from zyBV compelling. The paper does not clearly articulate why human judgment should be preferred over machine-side metrics for assessing explanation quality, especially given prior work showing that explanations can appear plausible to humans without faithfully reflecting the model's reasoning. My own view is that work would greatly benefit from focusing on a restricted set of applications. In this case, I see interesting ways to use this framework with reasoning models (e.g., to reject based on sycophantic reasoning from LLMs or to teach them *how* to reason). To be clear, my decision is based on reviews but I wanted to add my 2 cents since I think there is an opportunity here to reap benefits from a far more interesting class of applications.

**Reviewer Concerns:**

1. Rationale for Human Judgment (unclear motivation for prioritizing user-perceived plausibility over machine-side faithfulness metrics, given prior work showing these can diverge) (zyBV, 4xMk)

2. LLM-Simulated Labels (benchmark experiments use LLMs to generate "human" quality judgments, raising concerns about whether results reflect genuine human reasoning or LLM artifacts) (fLkm, 4xMk, zyBV)

**Reviewer Scores:**

I would have expected each review to give the standard 1-2 point boosts across the board. Leading to a (4,4,6,8). This would have been a borderline with AC making the call. The original AC would have most likely deferred to the judgement of 4xMk and the paper would have been a reject.

---

### Decision · Program_Chairs · 2026-01-26

Reject